# Macroalgal metabolism and lateral carbon flows can create significant carbon sinks

Kenta Watanabe[1], Goro Yoshida[2], Masakazu Hori[2], Yu Umezawa[3], Hirotada Moki[1], and Tomohiro Kuwae[1]

[1]Coastal and Estuarine Environment Research Group, Port and Airport Research Institute, 3-1-1 Nagase, Yokosuka 239-0826, Japan
[2]National Research Institute of Fisheries and Environment of Inland Sea, Japan Fisheries Research and Education Agency, 2-17-5 Maruishi, Hatsukaichi 739-0452, Japan
[3]Department of Environmental Science on Biosphere, Tokyo University of Agriculture and Technology, 3-5-8 Saiwai-cho, Fuchu, Tokyo 183-8509, Japan

*Correspondence to*: Kenta Watanabe (watanabe-ke@p.mpat.go.jp)

**Abstract.** Macroalgal beds have drawn attention as one of the vegetated coastal ecosystems that act as atmospheric $CO_2$ sinks. Although macroalgal metabolism as well as inorganic and organic carbon flows are important pathways for $CO_2$ uptake by macroalgal beds, the relationships between macroalgal metabolism and associated carbon flows are still poorly understood. In the present study, we investigated carbon flows, including air–water $CO_2$ exchange and budgets of dissolved inorganic carbon, total alkalinity, and dissolved organic carbon (DOC), in a temperate macroalgal bed during the productive months of the year. To assess the key mechanisms responsible for atmospheric $CO_2$ uptake by the macroalgal bed, we estimated macroalgal metabolism and lateral carbon flows (i.e., carbon exchanges between the macroalgal bed and the offshore) by using field measurements of carbon species, a field-bag method, a degradation experiment, and mass-balance modelling in a temperate *Sargassum* bed over a diurnal cycle. Our results showed that macroalgal metabolism and lateral carbon flows driven by water exchange affected air–water $CO_2$ exchange in the macroalgal bed and the surrounding waters. Macroalgal metabolism caused overlying waters to contain low concentrations of $CO_2$ and high concentrations of DOC that were efficiently exported offshore from the macroalgal bed. These results indicate that the exported water can potentially lower $CO_2$ concentrations in the offshore surface water and enhance atmospheric $CO_2$ uptake. Furthermore, the *Sargassum* bed exported 6–35 % of the macroalgal net community production (NCP; 302–1378 mmol-C $m^{-2}$ $d^{-1}$) as DOC to the offshore. The results of degradation experiments showed that 56–78 % of macroalgal DOC was refractory DOC (RDOC) that persisted for 150 days; thus, the *Sargassum* bed exported 5–20 % of the macroalgal NCP as RDOC. Our findings suggest that macroalgal beds in habitats associated with high water exchange rates can create significant $CO_2$ sinks around them and export a substantial amount of DOC to offshore areas.

## 1 Introduction

Vegetated coastal ecosystems provide a variety of ecosystem functions that support diverse biological communities and biogeochemical processes. Recent recognition of the carbon sequestration function of these ecosystems has led to the development of Blue Carbon strategies for mitigating the adverse effects of global climate change via conservation and restoration of these ecosystems (Nellemann et al., 2009; Duarte et al., 2013; Macreadie et al., 2019).

Carbon flows that sequester atmospheric $CO_2$ in marine ecosystems over timescales of at least several decades are crucial for the mitigation of climate change (McLeod et al., 2011; Macreadie et al., 2019). Organic carbon burial in sediments is one of the most important pathways to sequester carbon for a long time (Nellemann et al., 2009; Miyajima et al., 2019). Evaluation of the carbon sequestration function of vegetated coastal ecosystems has thus far been focused on saltmarshes, seagrasses, and mangroves, which develop their own organic-rich sediments (Macreadie et al., 2019). In contrast, beds of macroalgae have been assumed to have limited capacity to sequester carbon because they generally settle on hard strata such as rocks and artificial structures (Krause-Jensen et al., 2018). Organic matter produced by macroalgae shows variable lability but it is generally more labile than that produced by vascular plants (Trevathan-Tackett et al., 2015) and hence is more efficiently utilized by consumers and decomposers (Duarte, 1995). However, macroalgal beds are estimated to be the most extensive vegetated coastal habitats (3.5 million km$^2$) in the global ocean, and their global net primary production (1521 Tg-C yr$^{-1}$) is larger than that of other vegetated coastal habitats (Krause-Jensen and Duarte, 2016; Duarte, 2017; Raven, 2018). Macroalgal beds therefore have the potential to regulate carbon dynamics in coastal ecosystems.

Other processes in addition to organic carbon burial in on-site sediments must exist for macroalgae to contribute to atmospheric $CO_2$ sequestration. Recent studies have proposed that a large fraction of macroalgal production is exported to other vegetated coastal ecosystems, shelves, and the deep sea, where organic carbon derived from macroalgae can be stored in sediments and the water column for a long time (Krause-Jensen and Duarte, 2016; Krause-Jensen et al., 2018; Queirós et al., 2019).

The export and persistence of macroalgal dissolved organic carbon (DOC) has been proposed to be principal processes of macroalgal carbon sequestration, but more empirical support is needed to quantify this carbon flow. Macroalgal beds export about 43 % of their production as DOC and particulate organic carbon (POC) (Krause-Jensen and Duarte, 2016). A first-order estimate has suggested that 33 % of the flux of DOC derived from macroalgae is exported to depths below the mixed layer, where it contributes to carbon sequestration (Bauer and Druffel, 1998; Krause-Jensen and Duarte, 2016). Because the proportion of exported carbon that persists for a long time is estimated to be higher in DOC (33%) than in POC (15 %) (Krause-Jensen and Duarte, 2016), DOC production, export, and degradation are believed to be significant processes for carbon sequestration. Although the production of refractory DOC by macroalgae is one of the important factors that impact carbon sequestration, there are few relevant data (e.g., Wada et al., 2008; Wada and Hama, 2013). The long residence time of refractory DOC in the water column increases the probability that it reaches depths below the mixed layer.

Even though macroalgal beds perform a significant function by assimilating organic carbon, the chemical kinetics of the carbonate system in the water column could cause them to be net $CO_2$ emitters via air–water $CO_2$ exchange. The dissolved constituents of the carbonate system must therefore be assessed to quantify the effect of community metabolism on air–water $CO_2$ exchange (Macreadie et al., 2019; Tokoro et al., 2019). The high rates of macroalgal photosynthesis and respiration change dissolved inorganic carbon (DIC) concentrations. Calcification and dissolution of associated organisms modify the total alkalinity (TAlk) and DIC. Physical parameters and the balance of the carbonate system decide the magnitude of the air–water $CO_2$ exchange (Tokoro et al., 2019). Indeed, some previous studies have shown that macroalgal beds act as sinks for atmospheric $CO_2$ (Delille et al., 2009; Ikawa and Oechel, 2015; Koweek et al., 2017) and contribute substantially to global carbon fluxes (Smith, 1981; Krause-Jensen and Duarte, 2016). Macroalgal metabolism regulates diurnal and temporal variations in carbonate chemistry and affects calcification by calcifiers in macroalgal beds (Middelboe and Hansen, 2007; Krause-Jensen et al., 2015, 2016; Duarte and Krause-Jensen, 2018; Wahl et al., 2018). However, there is limited field evidence for how the effects of macroalgal metabolism on the carbonate system extend to adjacent water bodies.

Despite the importance of dissolved carbon flows as $CO_2$ sequestration pathways, little attention has been paid to assessing the related carbon budgets in macroalgal beds. In this study, we assessed carbon flows, including air–water $CO_2$ exchange and changes of DIC, TAlk, and DOC, in a temperate macroalgal bed during productive periods (winter). To quantify macroalgal metabolism and dissolved carbon flows, we used a field-bag method, a degradation experiment, and mass balance modelling. In the present study, we focused on *Sargassum* beds because they are one of the dominant macroalgal habitats in both temperate and tropical regions (e.g., Fulton et al., 2019; Yoshida et al., 2019). Our goals were to quantify the contribution of macroalgal beds to atmospheric $CO_2$ uptake and to investigate the responsible mechanisms on a daily timescale.

## 2 Materials and methods

### 2.1 Study site and sample collection

This study was conducted in the coastal waters of Heigun Island (33°46'1.7"N, 132°15'24.3"E) in the western Seto Inland Sea of Japan (Fig. 1). The macroalgal bed at the study site is dominated by *Sargassum* algae (Figs. S1 and S2 in the Supplement). The surface area of the macroalgal bed is 1.44 ha, and the macroalgal habitat is located at depths shallower than 5 m (mean depth, 2.0 m). There is no significant freshwater input from the island. The study site is characterized by a relatively high tidal amplitude (<4 m), and it is adjacent to a deep strait (~60 m).

Field surveys were conducted in February and March of 2019 in the macroalgal bed and the adjacent water bodies to take into account the temporal variations of biotic and abiotic conditions. Winter, including the months of February and March, is the most productive period of *Sargassum* algae around this study site (Yoshida et al., 2001). Surface water samples for analyses of DIC, TAlk, and DOC were collected from a research vessel three times (10:00, 13:00, and 16:00) during the daytime (approximately from 7:00 to 17:00) during both surveys at five stations (H1–H5) (Fig. 1). Four stations (H1–H4)

inside the macroalgal bed were chosen at equal intervals between the ends of the bed to assess average conditions. Station
H5 was established at an offshore site. Samples for DIC and TAlk were dispensed into 250-mL Schott Duran bottles and
preserved with mercuric chloride (200 µL per bottle) to prevent DIC changes due to biological activity. Water samples for
DOC analysis were filtered through 0.2-µm polytetrafluoroethylene filters (DISMIC–25HP; Advantec, Durham, NC, USA)
into precombusted (450 °C for 2 h) 50-mL glass vials and frozen at −20 °C until analysis. At each station, the salinity,
temperature, and chlorophyll fluorescence of the surface water were recorded with a RINKO-Profiler (ASTD102, JFE
Advantech, Nishinomiya, Japan).
Field bag experiments (e.g., Wada et al., 2007; Towle and Pearse, 1973) were conducted to quantify the changes of DIC,
TAlk, and DOC by macroalgae during one day in both February and March of 2019. We selected *Sargassum horneri* as the
subject species because sufficient amounts of *S. horneri* were present in a zone suitable for the experiments. The entire
thallus of an individual *S. horneri* was covered with a plastic bag containing ambient seawater collected in the macroalgal
bed. The open end of the bag was tied at the algal stipe by scuba divers. Triplicate transparent and dark bags were set up to
measure the changes of dissolved constituents due to macroalgal metabolism (Fig. S3 in the Supplement). To assess the
effect of phytoplankton, a set of transparent and dark bags were filled with ambient seawater that was collected in the
macroalgal bed but contained no macroalgae. These bags served as control bags. Water samples from the bags were
collected just after the start of the experiment and about 4 h later through a Tygon® tube by using a hand-held vacuum pump.
The collected water samples were preserved with mercuric chloride for the carbonate chemistry analysis and filtered through
the 0.2-µm filters for the DOC analysis (vide supra). After the experiments, the volume of seawater and the wet weight of the
macroalgae in each bag were measured. At the beginning and end of the experiments, the salinity, temperature, and
chlorophyll fluorescence of the surface water were recorded with a RINKO-Profiler (ASTD102, JFE Advantech).
Photosynthetic photon flux was measured with a photon flux sensor (DEFI-L, JFE Advantech) during the experiments.
The assessment of the biomass and species composition of the macroalgal bed that we studied was conducted in March
2019. Two 120-m transect lines were set from the shoreline to the edge of the macroalgal bed to document the biomass,
coverage, and species composition of the macroalgae (Fig. 1). To assess the coverage and species composition, 1 m × 1 m
quadrats were located at 10-m intervals along each transect. SCUBA divers quantified the apparent vegetation coverage and
species composition in each quadrat. Five quadrats (0.5 m × 0.5 m) were randomly located in the area dominated by
*Sargassum* algae along each transect to quantify the wet weight biomass (g WW) of the macroalgae. SCUBA divers
collected all macroalgae in each quadrat. The wet weight of the *Sargassum* algae and the other macroalgae were then
measured immediately.
**2.2 Degradation experiment**
To quantify the degradation rates of macroalgal DOC due to microbial activity and to estimate the refractory fraction of that
DOC, DOC samples for degradation experiments were obtained after the field bag experiments. Water samples were
collected from each transparent bag of macroalgae and control. The samples were filtered through precombusted (450 °C for
2 h) glass fibre filters (0.7-µm pore size; GF/F, Whatman, Maidstone, Kent, UK) under reduced pressure. We assumed that
GF/F filters would allow the passage of a significant fraction of free-living bacteria into the experimental samples (e.g.,
Wada et al., 2008; Bauer and Bianchi, 2011; Kubo et al., 2015).
The 40-mL filtrates were transferred into precombusted (450 °C for 2 h) 100-mL glass vials sealed with rubber and
aluminium caps. The 60-mL headspace in each glass bottle contained about 540 µmol oxygen, which was sufficient to
support the aerobic microbial degradation of DOC (~220 µmol) in each bottle if 1 mol of oxygen was consumed by the
mineralization of 1 mol of DOC into $CO_2$. The degradation experiments were conducted based on a total of six incubations
(0, 3, 10, 30, 90, and 150 days) per field survey. Triplicate bottles were used for each incubation. The experimental samples
were stored at room temperature (22 °C) in total darkness until analysis. In the present study, we used room temperature for
both samples to evaluate the quality of the organic matter. After incubation, the samples were filtered through 0.2-µm
polytetrafluoroethylene filters (DISMIC–25HP; Advantec) into precombusted (450 °C for 2 h) 100-mL glass vials and
frozen at −20 °C until analysis.
In this study, the concentration of refractory DOC (RDOC) was defined as the concentration of DOC remaining after 150
days, and the concentration of DOC derived from macroalgae ($DOC_M$) was equated to the difference between the DOC
concentration in the macroalgae bag and the DOC concentration in the control bag ($DOC_C$). Degradation rates ($k$) were
calculated by a first-order exponential decay model as follows:
$$DOC_{M(t)} = DOC_{M(0)} \times e^{-kt}$$
(1)

where $DOC_{M(t)}$ is the amount of $DOC_M$ remaining at time $t$ (day), and $k$ is the degradation rate ($day^{-1}$).

## 2.3 Sample analyses

The DIC concentration and TAlk were determined with a batch-sample analyser (ATT-05 and ATT-15; Kimoto Electric,
Osaka, Japan) according to Tokoro et al. (2014). The analytical precision of the system, based on the standard deviation of
multiple reference replicates, was normally within ±2 µmol $L^{-1}$ for DIC and TAlk.
DOC concentrations were measured at least in triplicate with a total organic carbon analyser (TOC-L; Shimadzu, Kyoto,
Japan) as non-purgeable organic carbon according to Ogawa et al. (1999). Potassium hydrogen phthalate (Wako Pure
Industries, Osaka, Japan) adjusted to three concentrations (83, 166, and 332 µM) was used as a standard for the measurement.
The coefficient of variation of the analyses was less than 2 %.

## 2.4 Metabolic parameters

Net community production (NCP), gross community production (GCP), community respiration (R), community
calcification (CC), and net DOC release (NDR) were determined from the changes in DIC, TAlk, and DOC of the field bag
experiments. These metabolic parameters were determined for both control and macroalgae as follows:
$$\text{Control NCP (μmol-C L}^{-1}\text{ h}^{-1}\text{)} = -\frac{\Delta DIC_L - 0.5 \times \Delta TAlk_L}{T} \qquad (2)$$
$$\text{Control R (μmol-C L}^{-1}\text{ h}^{-1}\text{)} = \frac{\Delta DIC_D - 0.5 \times \Delta TAlk_D}{T} \qquad (3)$$
$$\text{Control GCP (μmol-C L}^{-1}\text{ h}^{-1}\text{)} = \text{Control NCP} + \text{Control R} \qquad (4)$$
$$\text{Control CC (μmol-C L}^{-1}\text{ h}^{-1}\text{)} = -\frac{0.5 \times \Delta TAlk}{T} \qquad (5)$$
$$\text{Control NDR (μmol-C L}^{-1}\text{ h}^{-1}\text{)} = \frac{\Delta DOC}{T} \qquad (6)$$
$$\text{Macroalgal NCP (μmol-C g WW}^{-1}\text{ h}^{-1}\text{)} = \frac{V}{B} \times \left( -\frac{\Delta DIC_L - 0.5 \times \Delta TAlk_L}{T} - \text{Control NCP} \right) \qquad (7)$$
$$\text{Macroalgal R (μmol-C g WW}^{-1}\text{ h}^{-1}\text{)} = \frac{V}{B} \times \left( \frac{\Delta DIC_D - 0.5 \times \Delta TAlk_D}{T} - \text{Control R} \right) \qquad (8)$$
$$\text{Macroalgal GCP (μmol-C g WW}^{-1}\text{ h}^{-1}\text{)} = \text{Macroalgal NCP} + \text{Macroalgal R} \qquad (9)$$
$$\text{Macroalgal CC (μmol-C g WW}^{-1}\text{ h}^{-1}\text{)} = \frac{V}{B} \times \left( -\frac{0.5 \times \Delta TAlk}{T} - \text{Control CC} \right) \qquad (10)$$
$$\text{Macroalgal NDR (μmol-C g WW}^{-1}\text{ h}^{-1}\text{)} = \frac{V}{B} \times \left( \frac{\Delta DOC}{T} - \text{Control NDR} \right) \qquad (11)$$
In both the control and macroalgal field bag experiments, $\Delta DIC$, $\Delta TAlk$, and $\Delta DOC$ were equated to the final
concentrations minus the initial concentrations. The subscripts L and D indicate transparent (i.e., light) and dark bags,
respectively. The variables $V$, $B$, and $T$ are the volume of seawater (L), the wet weight of the macroalgae (g WW), and the
incubation time (h) in each bag, respectively. The CC and NDR rates were calculated for the daytime and night-time
separately by using the data from the light and dark experiments, respectively. The metabolic parameters were converted to
daily areal rates (mmol-C m$^{-2}$ d$^{-1}$) by using the mean macroalgal biomass, the mean water depth, the lengths of the
photoperiods, and the results of both daytime and night-time experiments. The photoperiod was defined as the time interval
between sunrise and sunset; photoperiods were obtained from Automated Meteorological Data Acquisition System
(AMeDAS) data provided by the Japan Meteorological Agency (available at: https://www.jma.go.jp, 2020).

## 2.5 Air–water CO₂ flux

The air–water $CO_2$ flux ($FCO_2$) was determined by using the bulk formula method. The equation for the method is as follows:

$$FCO_2 = -K \times S \times (fCO_{2water} - fCO_{2air})$$ (12)

where $fCO_2$ is fugacity. The gas transfer velocity ($K$) was determined from empirical relationships between $K$ and the wind speed above the surface of the water (e.g., Wanninkhof, 1992; McGillis et al., 2001). $S$ is the $CO_2$ solubility in the water. A positive $FCO_2$ value indicates $CO_2$ uptake from the air to the water. Here we used the following empirical equation to estimate $K$ (Wanninkhof, 1992):

$$K = 0.39 \times U_{10}{}^2 \times \left(\frac{Sc}{660}\right)^{-0.5}$$ (13)

where $U_{10}$ is the wind speed at a height of 10 m above the water surface. We determined $U_{10}$ by assuming that there was a logarithmic relationship between wind speed, height, and the roughness of the water surface (Kondo, 2000). Wind speed was obtained from AMeDAS data provided by the Japan Meteorological Agency and was measured about 10 km away at Agenosho (altitude: 6.5 m) (available at: https://www.jma.go.jp, 2020). The Schmidt number ($Sc$) was determined from the water temperature and salinity of the water surface.

The solubility ($S$) of $CO_2$ is a function of water temperature and salinity (Weiss, 1974). $fCO_{2water}$ and $fCO_{2air}$ are the fugacities of $CO_2$ in water and air, respectively. The values of $fCO_{2water}$ were estimated with the CO2SYS program (Lewis and Wallace, 1998) and the TAlk and DIC of the water samples (Zeebe and Wolf-Gladrow, 2001). The average salinity and water temperature were used to calculate $fCO_{2water}$ in each survey. We used the averaged $fCO_{2air}$ (410 µatm) measured with a $CO_2$ analyser (CO2-09; Kimoto Electric, Osaka, Japan).

## 2.6 Mass balance modelling

We simulated the diurnal changes and budgets of the carbonate system and DOC in the macroalgal bed by using mass balance models (Fig. 2). The mass balance models of the macroalgal bed simulated a hypothetical average macroalgal bed covering an area of 1 m². The average depth of the hypothetical macroalgal bed was the same as that of the macroalgal bed at the study site (2.0 m), and the tide was simulated by changing the water height in synchrony with the observed tide. We used the average biomass of *Sargassum* algae obtained from the field survey in the mass balance models. This modelling was conducted solely for the macroalgal bed, and the observed values of the offshore site (H5) were used as the boundary conditions for carbon inflowing into the macroalgal bed.

Time course changes in the concentrations of DIC, TAlk, and DOC (µmol L$^{-1}$) in the macroalgal bed were calculated at
hourly time intervals (Fig. 2). The duration of the simulation was 24 h beginning at sunrise of the survey day. Each
concentration at time step ($t$) was calculated from the concentration at time step ($t - 1$) as follows:
$$DIC_{(t)} = \left(DIC_{(t-1)} - GCP + R - CC - FCO_2\right) \times \left(1 - EX_{(t)}\right) + DIC_O \times EX_{(t)} \tag{14}$$
$$TAlk_{(t)} = \left(TAlk_{(t-1)} - 2CC\right) \times \left(1 - EX_{(t)}\right) + TAlk_O \times EX_{(t)} \tag{15}$$
$$DOC_{(t)} = \left(DOC_{(t-1)} + NDR\right) \times \left(1 - EX_{(t)}\right) + DOC_O \times EX_{(t)} \tag{16}$$
Metabolic parameters (GCP, R, CC, and NDR) were determined from changes in DIC, TAlk, and DOC measured in the
field bag experiments (Fig. 2 and Table S1 in the Supplement). These metabolic parameters were calculated as the sum of
the contributions from both macroalgae and phytoplankton. The parameters $DIC_O$, $TAlk_O$, and $DOC_O$ in Eqs. (14–16) are the
mean values of DIC, TAlk, and DOC, respectively, at the offshore station (H5), and the initial values in the simulation were
equated to those values. Namely, $DIC_{(0)}$, $Talk_{(0)}$, and $DOC_{(0)}$ were equated to $DIC_O$, $TAlk_O$, and $DOC_O$, respectively. We
assumed that there was no biogeochemical exchange between the bottom substrate and water. In the simulation, we assumed
that the metabolic parameters (GCP, R, CC, and NDR) of *S. horneri* were applied to the entire macroalgal bed and used
different metabolic parameters for day and night. $EX$ ($0 \leq EX \leq 1$), the hourly water exchange rate, was defined as follows:
$$EX_{(t)} = EX_{tide(t)} + EX_r \tag{17}$$
$$EX_{tide(t)} = \begin{cases} \dfrac{H_{(t)} - H_{(t-1)}}{H_{(t)}} & \left(H_{(t)} \geq H_{(t-1)}\right) \\ 0 & \left(H_{(t)} < H_{(t-1)}\right) \end{cases} \tag{18}$$
$EX_{tide}$ indicates the water exchange rate due to tidal change. $EX_{tide}$ was estimated from the changes of water height ($H$) and
was positive during the flood tide and zero during the ebb tide. $EX_r$ was defined as the residual exchange rate due to factors
other than tidal exchange (e.g., wind-driven water exchange and coastal currents). The value of $EX_r$ was chosen so as to
minimize the root mean square error (RMSE) of the modelled values versus the observed values. RMSEs were calculated for
the $z$-scores of DIC, TAlk, DOC, and fCO$_2$, which were equated to the differences between the modelled values and the
means of the observed values divided by the standard deviations of the observed values. The value of $EX_r$ that minimized the
averaged RMSEs for these four parameters was determined for each survey. This model fitting was performed using the
daytime data. The estimated $EX_r$ was applied throughout the diurnal cycle on the assumption that $EX_r$ was comparable during
the day and night. We ran two different model scenarios, one with and the other without water exchange (i.e., $EX$).
The budgets of DIC, TAlk, and DOC were calculated as the net gain or loss of each constituent due to water exchange.
The changes in fCO$_2$, which were estimated by using chemical equilibrium relationships and the TAlk and DIC of the water
samples (Lewis and Wallace, 1998; Zeebe and Wolf-Gladrow, 2001), were used to calculate $FCO_2$. The average salinity and
water temperature were used to calculate $fCO_2$ in each survey.

**2.7 Statistical analyses**

Statistical analyses were performed by using R statistical packages (R Core Team, 2019). We used a Welch's two-sample $t$-
test to determine whether there were differences in salinity, DIC, TAlk, $fCO_2$, and DOC between the macroalgal bed and the
offshore site and to detect the differences between the initial and final concentrations of DOC during degradation
experiments.

**3 Results**

**3.1 Carbonate system and DOC in the macroalgal bed**

There were no differences in salinity and TAlk between the macroalgal bed ($n = 12$) and the offshore site ($n = 3$) in either
February or March (Welch's two-sample $t$-test, $p > 0.05$) (Fig. 3 and Table S2 in the Supplement). The DIC concentration
was significantly lower in the macroalgal bed ($1964 \pm 22$ µmol $L^{-1}$) than at the offshore site ($1991 \pm 1$ µmol $L^{-1}$) in February
($p = 0.002$) (Fig. 3 and Table S2 in the Supplement). In March, the variation of the DIC concentration was large ($1962 \pm 43$
µmol $L^{-1}$) in the macroalgal bed but was also significantly lower than at the offshore site ($1992 \pm 1$ µmol $L^{-1}$) ($p = 0.033$).
The $fCO_2$ values were significantly lower in the macroalgal bed than at the offshore site in both February ($p = 0.001$) and
March ($p = 0.025$) (Fig. 3 and Table S2 in the Supplement). The $fCO_2$ values in the macroalgal bed (February, $265 \pm 31$
µatm; March, $272 \pm 49$ µatm) and the offshore site (February, $305 \pm 3$ µatm; March, $309 \pm 1$ µatm) were lower than $fCO_{2air}$
(410 µatm). On average, the DOC concentrations were higher in the macroalgal bed than at the offshore site, but the
difference between them was significant only in March ($p = 0.010$) (Fig. 3 and Table S2 in the Supplement). $fCO_2$ was
strongly correlated with DIC in both February and March (Fig. 4). The homogeneous buffer factors ($\beta$), which were equated
to the slopes of log-log plots of $fCO_2$ versus DIC, were 10.81 and 9.36 in February and March, respectively.
Community carbon metabolism was calculated from the field-bag experiments (Table 1 and Table S1 in the Supplement).
The NCP of macroalgae was about four times higher in March (1378 mmol-C $m^{-2}$ $d^{-1}$) than in February (302 mmol-C $m^{-2}$
$d^{-1}$) (Table 1) and was considerably higher than that of phytoplankton (~22 mmol-C $m^{-2}$ $d^{-1}$). The net community
calcification (NCC) of macroalgae was positive during both months (11–21 mmol-C $m^{-2}$ $d^{-1}$), but the average carbon fluxes
due to NCC were one to two orders of magnitude lower than those associated with NCP. The net DOC release rates of
macroalgae were 107 mmol-C $m^{-2}$ $d^{-1}$ and 88 mmol-C $m^{-2}$ $d^{-1}$ in February and March, respectively. These values were
equivalent to about 35 % and 6 % of the NCP in February and March, respectively.

**3.2 Biomass and species composition of macroalgae**

The macroalgal bed was dominated by *Sargassum* algae (Fig. 5 and Figs. S1 and S2 in the Supplement). The biomass of *Sargassum* algae (mean: 4693 g WW m$^{-2}$) was higher than that of the other macroalgae (264 g WW m$^{-2}$) (Fig. 5). The coverage of *Sargassum* algae (~80 %) was also larger than that of the other macroalgae (~51 %).

**3.3 Degradation of DOC**

DOC concentrations collected from macroalgae bags decreased with time in both experiments (Welch's two-sample *t*-test, $p < 0.05$; Fig. 6). In contrast, the stability of DOC concentrations collected from control bags during the experiments ($p > 0.05$) suggested that DOC$_M$ gradually decreased with time. Refractory DOC$_M$ (RDOC$_M$) concentrations were 56 $\pm$ 4 % and 78 $\pm$ 27 % of initial DOC$_M$ concentrations in February and March, respectively (Fig. 6c). The degradation rate (*k*) for 150-day incubations was higher in February (0.0044 d$^{-1}$) than in March (0.0021 d$^{-1}$).

**3.4 Carbon budgets estimated using mass balance models**

The mass balance models simulated the temporal changes of carbonate chemistry and DOC concentrations for the two model scenarios, that is, with and without considering water exchange (Fig. 3). The RMSEs of the *z*-scores of the best-fitting models considering water exchange (mean: February, 0.56; March, 0.91) were lower than those assuming that water exchange was zero (mean: February, 3.77; March, 3.10) (Table 2). The fitted model that took into consideration *EX* improved the RMSEs of the *z*-scores of all parameters in February. In March, the RMSEs of the *z*-scores of DIC and fCO$_2$ were improved by the model fitting, but those of DOC and TAlk showed little or no improvement (Table 2). The estimated $EX_r$ values were 39 % and 42 % in February and March, respectively (Table 3). The $EX_r$ rates were the main components of the hourly water exchange rates (the sums of $EX_{tide}$ and $EX_r$), which were estimated to be 39–52 % and 42–68 % in February and March, respectively (Fig. 3 and Table 3).

DIC concentrations were decreased in the daytime by primary production (Fig. 3a, f). TAlk values in the macroalgal bed were stable and very similar to the TAlk values of the offshore seawater (Fig. 3b, g). The fCO$_2$ decreased during the daytime because of the concurrent decrease of the DIC concentration (Fig. 3c, h). DOC concentrations in the macroalgal bed exceeded those at the offshore site during the daytime (Fig. 3d, i).

DOC was exported offshore from the macroalgal bed (Fig. 7). The areal effluxes of DOC (February: 125 mmol-C m$^{-2}$ d$^{-1}$, March: 96 mmol-C m$^{-2}$ d$^{-1}$) were similar to the NDRs. The export fluxes of RDOC$_M$ were estimated to be 59 mmol-C m$^{-2}$ d$^{-1}$ and 67 mmol-C m$^{-2}$ d$^{-1}$ in February and March, respectively (Fig. 7). DIC budgets driven by water exchange indicated a net input of DIC from offshore to the macroalgal bed (Fig. 7 and Table 3). The areal influxes of DIC were 323 mmol-C m$^{-2}$ d$^{-1}$ and 1386 mmol-C m$^{-2}$ d$^{-1}$ in February and March, respectively. These fluxes were almost equivalent to the sum of NCP, NCC, and FCO$_2$ in the macroalgal bed (Fig. 7). The FCO$_2$ values showed that both the macroalgal bed and the

offshore site took up atmospheric $CO_2$ during these study periods. $FCO_2$ values were higher in the macroalgal bed than
offshore during both periods (Fig. 7 and Table 3).

## 4 Discussion

### 4.1 Refractory DOC release by macroalgae

Our results showed that the *Sargassum* bed released a large amount of DOC (Fig. 7). Most of the released DOC was
exported out of the macroalgal bed via water exchange during the day. The DOC release rates of *S. horneri* (18.7–22.8 μmol-
C g $WW^{-1}$ $d^{-1}$, Table S1 in the Supplement) were within the range of those reported for *Ecklonia* kelp (1.5–72.5 μmol-C g
$WW^{-1}$ $d^{-1}$, Wada et al., 2007), which were calculated by assuming that water content was 85 % of wet weight (Watanabe et
al., unpublished data). The fact that Wada et al. (2007) collected data over an entire year, whereas our data were collected
during only the most productive two months of the year, accounts for the difference in the variations of DOC release rates.
Previous studies have found that a substantial portion of production is released as DOC by kelps (18–62 %, Abdullah and
Fredriksen, 2004; Wada et al., 2007). Our results showed that Sargassum algae sometimes release a similar percentage of
production as DOC (February, 35 %; March; 6 %), and the percentages were very different between the two months, despite
the similarity of the DOC release rates (Fig. 7). DOC release rates by kelps have been shown to be correlated with irradiance,
but irradiance explained only 13 % of the variation of the DOC release rates (Reed et al., 2015). Time lags between light-
stimulated carbon assimilation and DOC release may explain some of the variation between irradiance and DOC release.
High-frequency time-series measurements may help to explain the daily variations of macroalgal carbon metabolism. In this
study, the reproducibility of the DOC mass balance model (i.e., the improvement of RMSEs) differed between the February
and March data sets (Fig. 3 and Table 2). Temporal and interspecific variations of DOC release rates may have caused this
difference.
Refractory organic carbon acts as a carbon reservoir in seawater (Hansell and Carlson, 2015) and is considered to be one
of the important contributors to carbon sequestration by coastal macrophytes (Maher and Eyre, 2010; Watanabe and Kuwae,
2015; Krause-Jensen and Duarte, 2016; Duarte and Krause-Jensen, 2017). Our results show that the *Sargassum* bed exported
5–20 % of the macroalgal NCP as RDOC that persisted for 150 days (Fig. 7). The fact that the degradation rates of
macroalgal DOC are lower than those of DOC released by phytoplankton ($k$ values, > 0.025 $d^{-1}$; Hama et al., 2004;
Kirchman et al., 1991) implies that macroalgal DOC is more biologically recalcitrant than DOC produced by phytoplankton
(Wada et al., 2008). Previous studies have suggested that macroalgae produce phenolic compounds such as phlorotannin that
are biologically recalcitrant (Swanson and Druehl, 2002; Wada and Hama, 2013; Powers et al., 2019). A thermogravimetric
approach has also shown that macroalgal thalli contain refractory compounds (Trevathan-Tackett et al., 2015), some of
which are released as the plant grows. These findings indicate that macroalgae release chemically recalcitrant DOC for
decomposers.
Wada et al. (2008) have estimated the turnover times of the DOC released by *Ecklonia* kelp, the reciprocals of the
degradation rates ($k$), to be 24–172 days (i.e., $k$ values of 0.0058–0.0407 $d^{-1}$) during 30-day incubations. In the present study,
the turnover times of DOC released by *S. horneri* were calculated to be 111–238 days (i.e., $k$ values for 30-day incubations
of 0.0042–0.0090 $d^{-1}$), longer than the turnover times of *Ecklonia* kelp. These findings indicate that the recalcitrance of
macroalgal DOC is variable and depends on the species and environmental conditions. The production of recalcitrant
macroalgae compounds such as phlorotannins varies among seasons, growth phases, and species (Steinberg, 1989; Kamiya
et al., 2010), and these variations may regulate seasonal and interspecific variations in the biological recalcitrance of
macroalgal DOC. Furthermore, degradation rates for 150-day incubations (0.0021–0.0044 $d^{-1}$; Fig. 6) were slower than those
for first 30-day incubations, indicating that relatively short duration degradation experiments may underestimate the long-
term persistence of OC (e.g., Trevathan-Tackett et al., 2020).
The microbial degradation of DOC is also affected by temperature, and high temperature stimulates DOC degradation
(e.g., Chen and Wangersky, 1996; Lønborg and Álvarez-Salgado, 2012). In this study, the microbial degradation rates of
DOC were potentially overestimated compared to in situ conditions because the incubation temperature for the degradation
experiments (22 °C) was higher than the in situ temperature (~13 °C, Table 1). The difference in the initial $DOC_M$
concentrations in the macroalgae bags between February and March may have been caused by the differences in the biomass
of macroalgae and volume of water in the experimental bags (Fig. 6a, b). Variations of DOC concentrations may affect
degradation rates via resource limitation of microbial activity (e.g., Arrieta et al., 2015). Understanding of the fate of
macroalgal DOC would be enhanced by the assessment of the physical and biochemical factors that regulate microbial
degradation of DOC. The rates of DOC degradation processes, which were not measured in this study (e.g., photochemical
degradation), might also be important in driving macroalgal DOC degradation (Wada et al., 2015).
Ogawa et al. (2001) have shown that marine bacteria take up labile organic matter (OM) such as glucose and convert it
into refractory OM. Some of the macroalgal DOC may be converted to refractory OC by microbes and persist in water for a
long time. Carbon flows through the microbial loop should be assessed as one of the fates of OM derived from macroalgal
beds.

### 4.2 $CO_2$ uptake and DIC budgets in the macroalgal bed

Atmospheric $CO_2$ uptake was affected by community metabolism and water exchange, which regulated the carbon budget in
the *Sargassum* algae-dominated macroalgal bed. Positive NCP values showed that the macroalgal bed acted as an
autotrophic system during the study periods. Macroalgal DIC uptake (i.e., NCP) accounted for >97 % of total NCP in this
system (Table 1); the rest was attributable to planktonic NCP. Biological uptake of DIC promoted atmospheric $CO_2$ uptake
by contributing to the decrease of DIC concentrations and $fCO_2$ during the day inside the macroalgal bed (Figs. 3 and 7).
Previous studies have shown that macroalgal primary production reduces DIC and $CO_2$ concentrations. For example,
DIC uptake by kelp reduces $fCO_2$ and thereby contributes to the uptake of atmospheric $CO_2$ inside kelp beds (Delille et al.,
2000, 2009; Koweek et al., 2017; Pfister et al., 2019). The aquaculture of macroalgal species such as the kelp *Laminaria*
*japonica* and the red algae *Gracilaria lemaneiformis* has also been shown to result in annual net uptake of $CO_2$ because of
active photosynthesis by the macroalgae (Jiang et al., 2013). In contrast, knowledge about in situ carbonate chemistry in beds
of *Sargassum* algae is limited (e.g., Tokoro et al., 2019). The present study, however, has shown that a bed of *Sargassum*
algae takes up atmospheric $CO_2$ over a diurnal cycle during productive periods of the year.
Our results showed that metabolism and water exchange regulated the diurnal variations in DIC and $fCO_2$ in the
macroalgal bed. Our mass balance model analyses suggested that the high rate of water inflow from the outside the bed
strongly affected DIC concentrations and $fCO_2$ in the macroalgal bed (Fig. 3a, f). The decrease of the DIC concentration of
the macroalgal bed was moderated by water exchange during the day. The high rate of water exchange reduced the
difference in $FCO_2$ between the inside and outside of the macroalgal beds (Fig. 7). Conversely, water characterized by low
DIC and $fCO_2$ values was efficiently exported from the macroalgal bed to the surrounding water (Fig. 7). Our findings
therefore suggested that macroalgal beds can create areas of adjacent water that serve as $CO_2$ sinks. Previous studies have
proposed that a canopy of the kelp genus *Macrocystis* dampens water exchange (Rosman et al., 2007), and the residence time
of water within kelp beds can reach several days (Jackson and Winant, 1983; Delille et al., 2009). In contrast, the exposed
side of a kelp bed is very much affected by the advection of offshore water (Koweek et al., 2017). Water exchange rates are
affected by the surface area of beds, canopy development, topography, and hydrological conditions.
The seasonality of the growth of macroalgae regulates the seasonal variations of carbonate chemistry and sink/source
behaviour (Delille et al., 2009; Koweek et al., 2017). Annual fluctuations of the surface area of kelp beds affect interannual
variations in air–water $CO_2$ fluxes in adjacent water bodies (Ikawa and Oechel, 2015). In the present study, we focused on
how daily carbon budgets were related to macroalgal metabolism and hydrological conditions during productive periods. The
biomass of *Sargassum* algae fluctuates seasonally and increases in winter (from November to April) around the present study
site (Yoshida et al., 2001). Future studies should assess the seasonal variability of carbonate chemistry in *Sargassum* beds.
The homogenous buffer factor ($\beta$) is a general and helpful tool that can be used to identify the main processes that affect
carbonate chemistry dynamics (e.g., Frankignoulle, 1994). Frankignoulle (1994) found the relationship $\beta = -7.02 + 0.186$
$\times$ %$C_{org}$, where %$C_{org}$ is the percent change of the DIC concentration due to photosynthesis and respiration. By using this
equation, we calculated the %$C_{org}$ to be 96 % and 88 % in February and March, respectively (Fig. 4). The results therefore
indicate that NCP was the main regulator of carbonate chemistry, and the contribution of NCC was relatively small. This
conclusion is consistent with the results of the field bag experiments (Table 1).
**4.3 Community metabolism in the macroalgal bed**
Macroalgal NCP values in the present study (302–1378 mmol-C $m^{-2}$ $d^{-1}$) were comparable to those in a sub-Arctic kelp bed
(~1250 mmol-C $m^{-2}$ $d^{-1}$; Delille et al., 2009) and to gross primary production in a *Macrocystis* kelp bed in California (~570
mmol-C $m^{-2}$ $d^{-1}$; Towle and Pearse, 1973; Jackson, 1987) and in an *Ecklonia* kelp bed (464 mmol-C $m^{-2}$ $d^{-1}$; Randall et al.,
2019); they were much larger than the NCP values in a calcareous macrophyte bed (19 mmol-C $m^{-2}$ $d^{-1}$; Bensoussan and
Gattuso, 2007), in temperate maerl beds (−38 mmol-C $m^{-2}$ $d^{-1}$; Martin et al., 2007), and on a coral reef dominated by green

and red algae ($-112$ to $61$ mmol-C m$^{-2}$ d$^{-1}$; Falter et al., 2001). The suppression of macroalgal R by low water temperatures during the productive winter can explain the relatively high NCP values observed at our study site (Table 1 and Table S1 in the Supplement). The macroalgal NCP value during March was four times higher than the value during February in the present study (Table 1). Irradiance, length of the photoperiod, and growth phase collectively control the temporal variations of macroalgal NCP. In the present study, both surveys were conducted during the productive period, but the difference in the averaged biomass per individual *S. horneri* used for the field bag experiments (February, 353 g WW; March 260 g WW) may indicate a difference in growth phase.

The relative growth rates (% d$^{-1}$) of *S. horneri* were calculated to be $1.1$–$7.3$ % d$^{-1}$ based on the ratio of growth (= NCP − NDR) to biomass (Table S1 in the Supplement). To calculate biomass, we assumed that the water content was 85 % of the wet weight and that carbon content was 30 % of the dry weight (Watanabe et al., unpublished data). These relative growth rates were comparable to estimates based on biomass changes of *S. horneri* (around 4 % d$^{-1}$, Gao and Hua, 1997; Choi et al., 2008) and *S. muticum* (~10 % d$^{-1}$, Pedersen et al., 2005).

The estimated uncertainties of NCC and NCP derived from the measurement precision of TAlk and DIC were ~13 mmol-C m$^{-2}$ d$^{-1}$ and ~26 mmol-C m$^{-2}$ d$^{-1}$, respectively. These uncertainties were similar to NCC values (macroalgae, $11$–$21$ mmol-C m$^{-2}$ d$^{-1}$; phytoplankton, $-12$–$3$ mmol-C m$^{-2}$ d$^{-1}$) and phytoplankton NCP values ($7$–$22$ mmol-C m$^{-2}$ d$^{-1}$) (Table 1). It is therefore difficult to discuss NCC values and phytoplankton NCP values quantitatively, but these values were substantially lower than macroalgal NCP values in this study. Increasing the incubation time in the field bag experiments should help to reduce these uncertainties.

## 4.4 Implications for the CO$_2$ sequestration function of macroalgae

Macroalgal beds are considered as potential carbon-donor sites in the context of Blue Carbon sequestration (Krause-Jensen et al., 2018). The release and subsequent export of particulate macroalgal carbon (e.g., entire thalli and fragments) via physical processes would contribute to CO$_2$ sequestration (Krause-Jensen and Duarte 2016; Filbee-Dexter et al., 2018; Pessarrodona et al., 2018; Kokubu et al., 2019; Pedersen et al., 2019) (Fig. 2). The export of recalcitrant DOC from macroalgal beds is also anticipated to be an important pathway of CO$_2$ sequestration (Wada and Hama, 2013; Barrón et al., 2014; Reed et al., 2015). A first-order assessment has suggested that almost 70 % of global macroalgal carbon sequestration is attributable to DOC export to depths below the mixed layer (Krause-Jensen and Duarte, 2016). Our results showed that a *Sargassum* bed released substantial amount of RDOC, which was rapidly exported from the habitat to the offshore. The maximum residence time of dissolved matter in the study's oceanographic basin is between $95$–$218$ days depending on the season (Balotro et al., 2002), indicating that macroalgal RDOC can be exported to the outside of the Seto Inland Sea and to depths below the mixed layer via vertical mixing.

The decrease in fCO$_2$ due to macroalgal DIC uptake directly controls the influx of atmospheric CO$_2$ into macroalgal habitats and the waters surrounding them. The present study showed that the metabolism of *Sargassum* algae mediated the

production of low-DIC and low-fCO$_2$ water, which was rapidly exported to outside the habitat. Because macroalgae
commonly inhabit rocky reefs facing the open ocean, macroalgal metabolism may affect a wide range of water bodies
surrounding rocky reef habitats (e.g., Ikawa and Oechel, 2015). The CO$_2$ sequestration function of macroalgae found in
habitats where macroalgae-affected water easily diffuses offshore has been overlooked.
Studies of the role of macroalgae in CO$_2$ sequestration should use field observations and coupled ecological-physical
models to assess the spatial spread and fate of DOC and low-fCO$_2$ waters derived from macroalgal habitats (Kuwae et al.,
2019; Macreadie et al., 2019). Because coastal primary producers other than macroalgae can also be a source of low-fCO$_2$
and high-DOC waters, separately analysing the fate of these waters would help shed light on the role of these ecosystems.
Seasonal variations in oceanographic and climatic conditions regulate the transport of waters affected by macroalgae. Such
studies will lead to a better understanding of the role of macroalgae in sequestering Blue Carbon and thereby mitigating
global climate change.

## 429 5 Conclusions

The present study showed that macroalgal metabolism and lateral carbon flows regulated carbon budgets and air–water CO$_2$
exchange in a temperate macroalgal bed and its surrounding water. Macroalgae took up DIC via photosynthesis and released
large amounts of DOC to the offshore waters adjacent to the bed. Hydrological water exchange enhanced the lateral carbon
flows and the spread of low-fCO$_2$ and high-DOC water mediated by macroalgal metabolism. Our findings suggest that
macroalgal beds have the potential to create areas of adjacent water that serve as CO$_2$ sinks. These results suggest the need
for future research to assess the areal extent and fate of macroalgae-mediated low-fCO$_2$ and high-DOC waters.

## 436 Data availability

The dataset used in this study can be obtained from Zenodo (http://doi.org/10.5281/zenodo.3715876; Watanabe et al., 2020).

## 438 Author contributions

KW, GY, MH, YU, and TK conceived the study. KW, GY, MH, HM, and TK collected the samples. KW and HM conducted
the laboratory analyses. KW and TK processed the data. KW and TK wrote the paper with substantial input from the other
authors.

## 442 Competing interests

The authors declare that they have no conflict of interest.

*Acknowledgements.* This study was funded in part by Grants-in-Aid for Scientific Research (KAKENHI) grant numbers JP18H04156, 19K20500, and 19K12295 from the Japan Society for the Promotion of Science. We thank A. Kajita, K. Manabe, and S. Sueyoshi for help in field observations and N. Umegaki, H. Kimishima, and R. Makino for chemical analyses. Finally, we thank A. Pessarrodona Silvestre and D. Krause-Jensen as reviewers for their useful comments that contributed to the development of the paper.

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

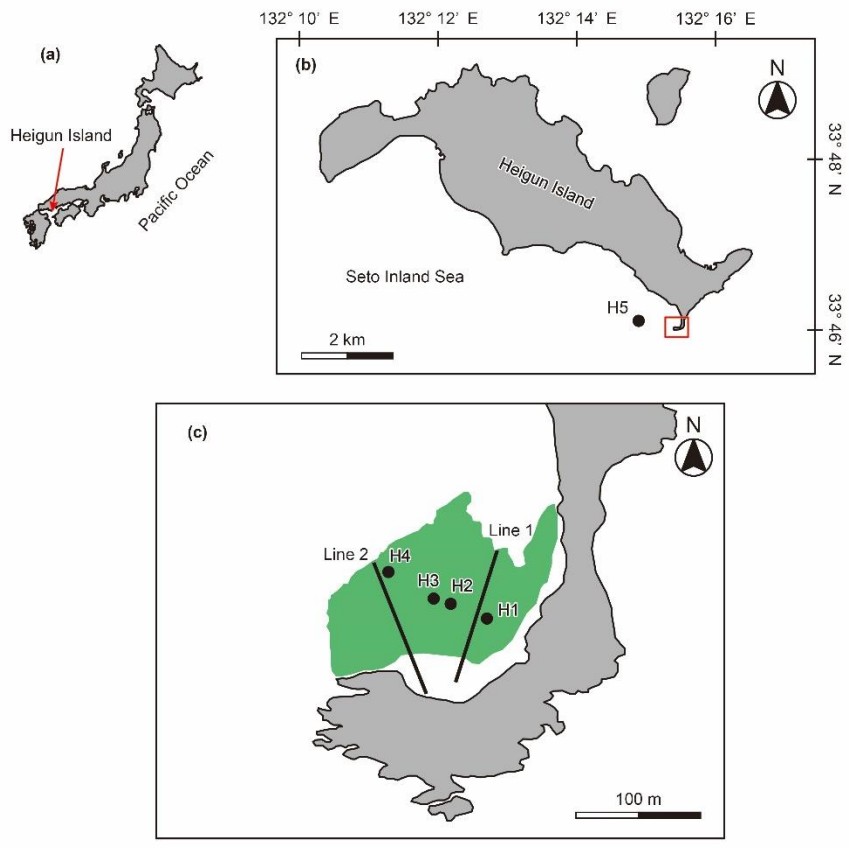


**Figure 1:** Maps of Heigun Island and the locations of sampling stations (H1–H5) and transect lines. Green shading indicates the area occupied by the macroalgal bed.


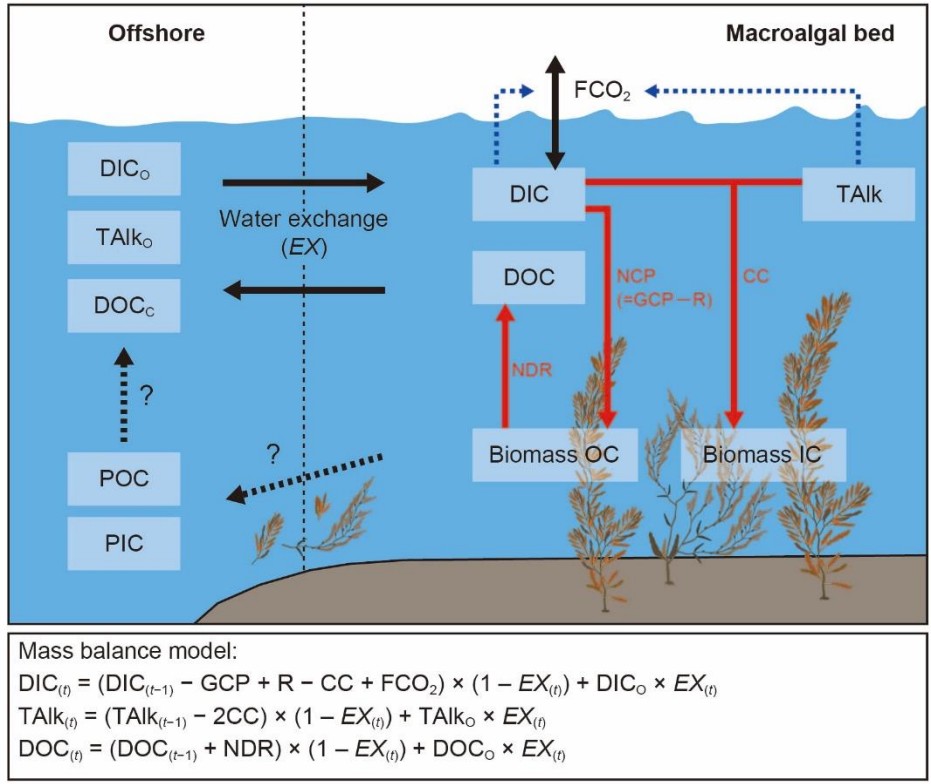

**Figure 2:** Schematic diagram of the different carbon pools and flows in and around macroalgal beds. Black arrows indicate
carbon flows between macroalgal beds and the outside of the system; black dashed arrows with question marks denote
carbon flows that were not evaluated in this study; red arrows indicate effects of community metabolism on carbon pools in
macroalgal beds. Blue dashed arrows indicate that dissolved inorganic carbon (DIC) and total alkalinity (TAlk) regulate air–
water $CO_2$ exchange fluxes ($FCO_2$). DIC concentrations in macroalgal beds are regulated by net community production
(NCP), community calcification (CC), $FCO_2$, and mixing with offshore DIC ($DIC_O$). NCP is calculated by subtracting
community respiration (R) from gross community production (GCP). TAlk in macroalgal beds is regulated by CC and
mixing with offshore TAlk ($TAlk_O$). Dissolved organic carbon (DOC) concentrations in macroalgal beds are regulated by net
DOC release (NDR) and mixing with offshore DOC ($DOC_O$). Organic carbon (OC) and inorganic carbon (IC) of macroalgal
biomass are produced by NCP and CC, respectively, and some of each is exported to the offshore in particulate form (POC
and PIC, respectively). Mass balance models simulated the diurnal changes and budgets of DIC, TAlk, and DOC in the
macroalgal bed at hourly time steps ($t$) in this study.


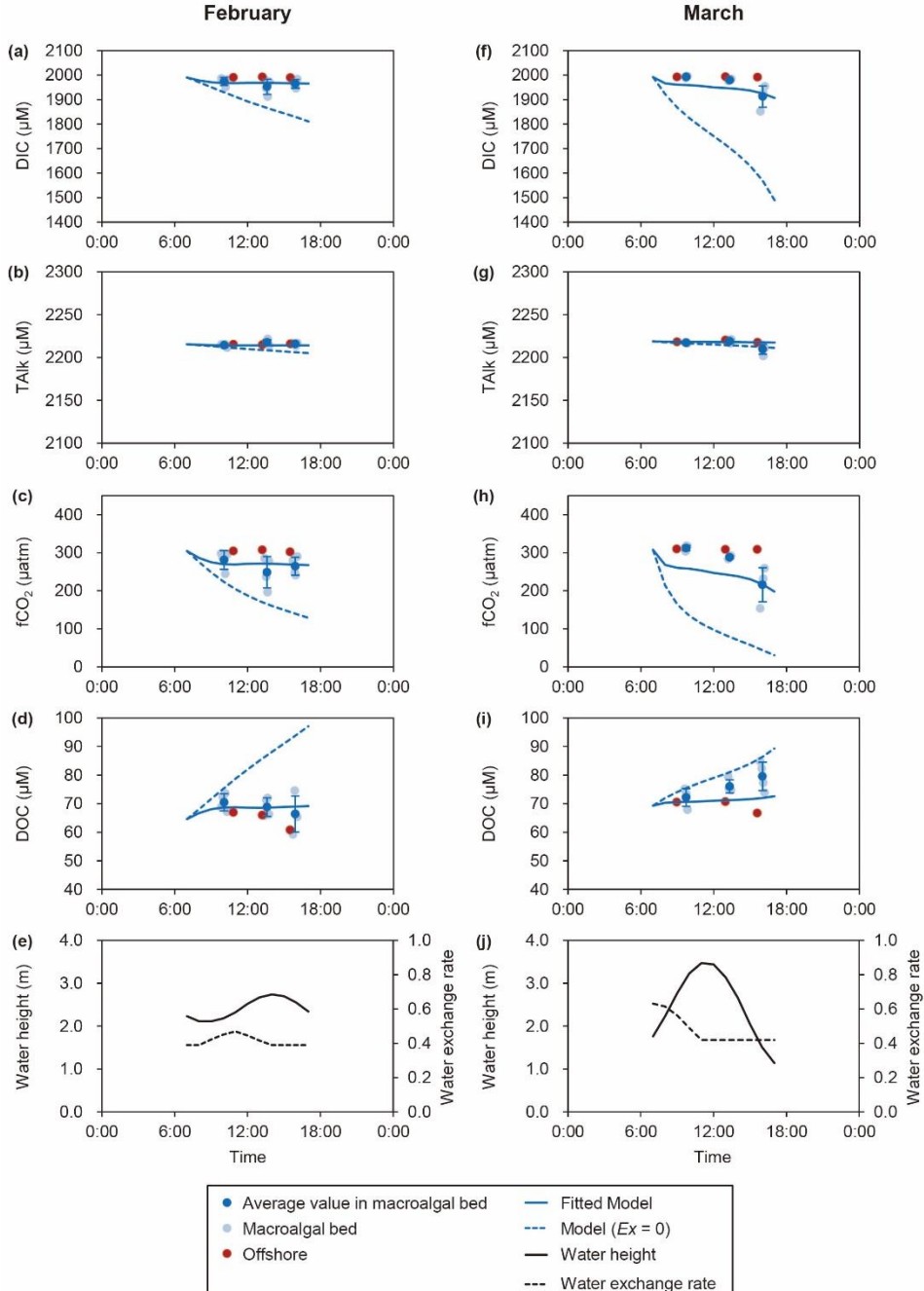

**Figure 3:** Temporal changes in dissolved inorganic carbon (DIC), total alkalinity (TAlk), fugacity of $CO_2$ ($fCO_2$), dissolved
organic carbon (DOC), water height, and water exchange rate (*EX*) in February (a–e) and March (f–j). Modelled values of
chemical parameters were estimated by using mass balance models. Error bars show standard deviations. Blue dashed lines
show the model results if the *EX* is zero. Details regarding observed values are provided in Table S2 in the Supplement.

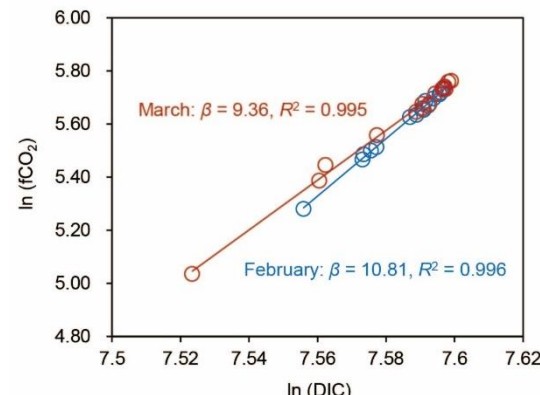

**Figure 4:** Plots of fugacity of $CO_2$ (f$CO_2$) versus dissolved inorganic carbon (DIC) and regression lines used to determine
the homogeneous buffer factors ($\beta$) as slopes.

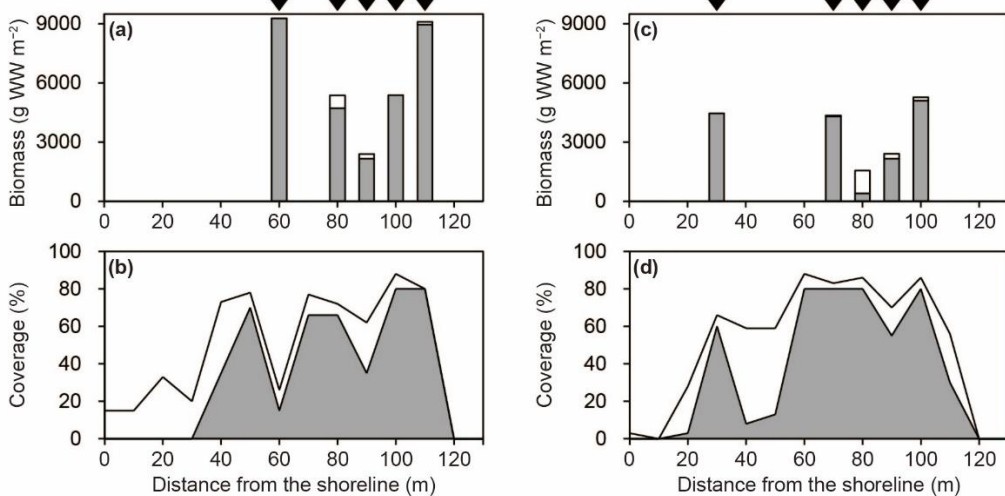

**Figure 5:** Biomass and coverage of macroalgae along transect line 1 (a, b) and line 2 (c, d) in March 2019. Grey and white shading indicate *Sargassum* algae and other macroalgae, respectively. Black arrows indicate sampling locations for macroalgal biomass.


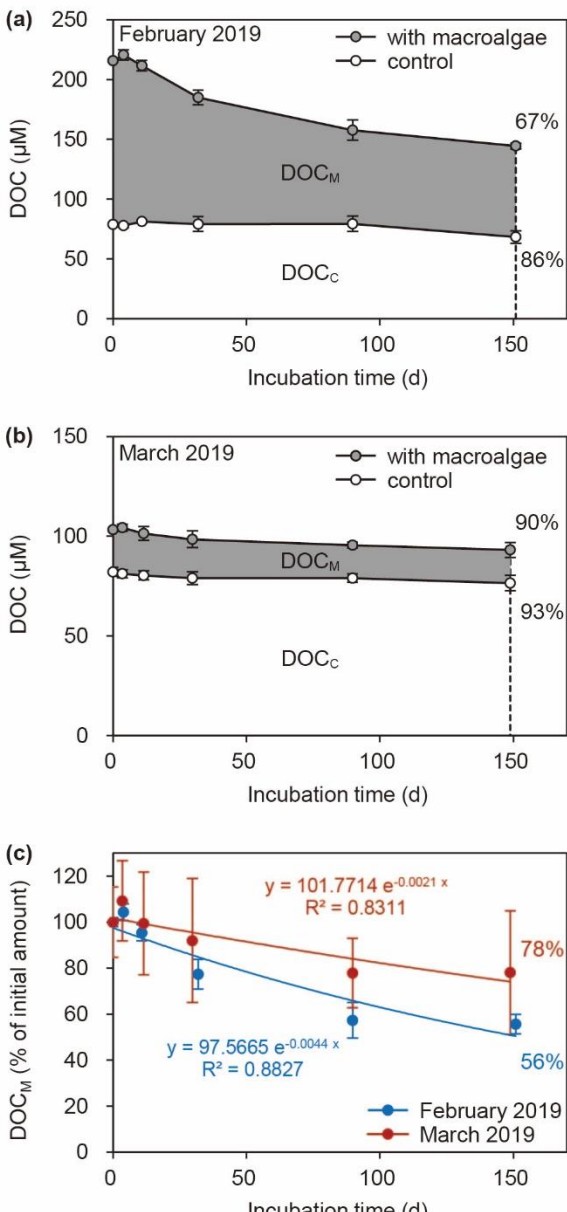

**Figure 6:** Time course of dissolved organic carbon (DOC) concentrations during the degradation experiments in (a)
February and (b) March and (c) percentage of DOC derived from macroalgae ($DOC_M$) during both experiments. Shading
indicates the concentration of $DOC_M$, which was equated to the difference between the DOC concentration in the macroalgae
bag and the DOC concentration in the control bag ($DOC_C$). Percentages in panels (a) and (b) are averaged final percentage of
$DOC_M$ and $DOC_C$ remaining in each treatment after 150 days.


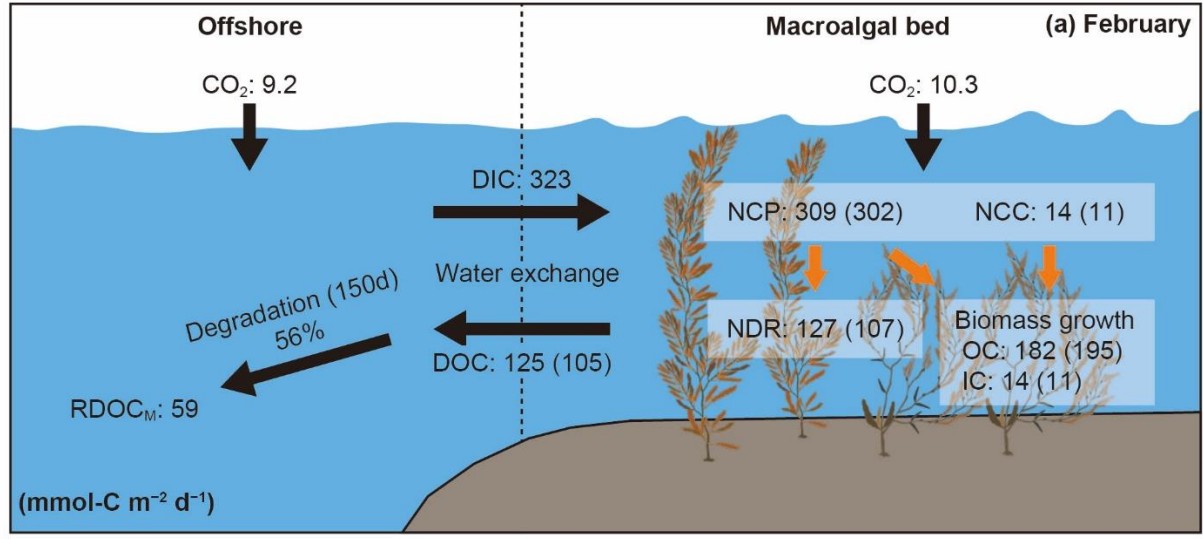

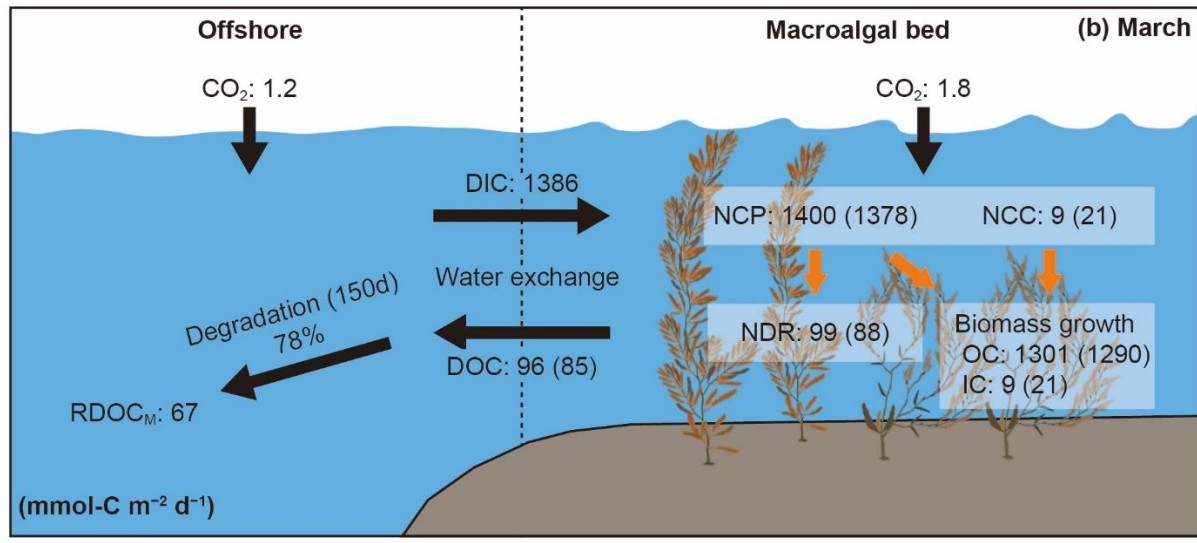

**Figure 7:** Carbon flows and community metabolism (NCP, net community production; NCC, net community calcification;
NDR, net DOC release) in the macroalgal bed. NCP, NCC, and NDR were calculated using the results of field bag
experiments (details are available in Table S1 in the Supplement). The carbon flows due solely to macroalgae are shown in
parentheses. Biomass growth in terms of organic carbon (OC) was calculated by subtracting NDR from NCP. Biomass
growth in terms of inorganic carbon (IC) was the same as NCC. DIC and DOC flows via water exchange were estimated by
mass balance modelling (details are available in Table 3). Community metabolism, biomass growth, and DOC outflow
indicate the sum of macroalgal and planktonic carbon flows. Carbon fluxes were calculated in units of millimoles per square
metre of the surface area of the macroalgal bed per day. RDOC$_M$ indicates refractory DOC released by macroalgae.


**Table 1:** Carbon metabolism, surface water temperature, photosynthetic photon flux, length of photoperiod, and chlorophyll fluorescence in February and March 2019. For macroalgae, means ± standard deviations are shown. Average water depth and biomass in the bed were used for calculating metabolic rates.


| Variables | Units | February 2019 | March 2019 |
|---|---|---|---|
| Macroalgae | | | |
| Net community production | mmol-C m$^{-2}$ d$^{-1}$ | 302 ± 130 | 1378 ± 660 |
| Gross community production | mmol-C m$^{-2}$ d$^{-1}$ | 572 ± 129 | 1637 ± 646 |
| Community respiration | mmol-C m$^{-2}$ d$^{-1}$ | 270 ± 18 | 259 ± 133 |
| Net DOC release | mmol-C m$^{-2}$ d$^{-1}$ | 107 ± 36 | 88 ± 37 |
| Net community calcification | mmol-C m$^{-2}$ d$^{-1}$ | 11 ± 7 | 21 ± 23 |
| Control (phytoplankton) | | | |
| Net community production | mmol-C m$^{-2}$ d$^{-1}$ | 7 | 22 |
| Gross community production | mmol-C m$^{-2}$ d$^{-1}$ | 22 | 4 |
| Community respiration | mmol-C m$^{-2}$ d$^{-1}$ | 15 | −18 |
| Net DOC release | mmol-C m$^{-2}$ d$^{-1}$ | 20 | 11 |
| Net community calcification | mmol-C m$^{-2}$ d$^{-1}$ | 3 | −12 |
| Surface water temperature | °C | 12.0 ± 0.2 | 12.4 ± 0.1 |
| Photosynthetic photon flux | μmol m$^{-2}$ s$^{-1}$ | 674 ± 595 | 1311 ± 202 |
| Length of photoperiod | h | 11 | 12.5 |
| Chlorophyll *a* concentration | μg L$^{-1}$ | 0.3 | 0.8 |




**Table 2:** Root mean square errors (RMSEs) for best-fitting models and models assuming that water exchange rate (*EX*) was zero. RMSEs were calculated for the *z*-scores of DIC, TAlk, DOC, and $fCO_2$ values, which were differences from the mean observed values divided by the standard deviations. The best-fitting model that minimized the averaged RMSEs for these four parameters was determined for each survey.


| Variables | February 2019 | | March 2019 | |
|---|---|---|---|---|
| | Best-fitting model | Model w/o *EX* | Best-fitting model | Model w/o *EX* |
| DIC | 0.41 | 4.26 | 0.69 | 6.64 |
| TAlk | 0.99 | 3.52 | 0.96 | 0.62 |
| DOC | 0.41 | 4.46 | 1.07 | 1.04 |
| $fCO_2$ | 0.44 | 2.85 | 0.91 | 4.10 |
| Mean | 0.56 | 3.77 | 0.91 | 3.10 |




**Table 3:** Water exchange rates ($EX_r$ and $EX_{tide}$), $FCO_2$, DIC exchange, and DOC exchange, which were estimated by using mass balance models. Carbon fluxes were calculated as millimoles per square metre of the surface area of the algal bed per day.


| Variables | Units | February 2019 | March 2019 |
|---|---|---|---|
| $EX_r$ | % h$^{-1}$ | 39 | 42 |
| $EX_{tide}$ | % h$^{-1}$ | 0–13 | 0–26 |
| $FCO_2$ in macroalgal bed | mmol-C m$^{-2}$ d$^{-1}$ | 10.3 | 1.8 |
| $FCO_2$ in offshore | mmol-C m$^{-2}$ d$^{-1}$ | 9.2 | 1.2 |
| DIC exchange | mmol-C m$^{-2}$ d$^{-1}$ | 323 | 1386 |
| DOC exchange | mmol-C m$^{-2}$ d$^{-1}$ | −125 | −96 |

