# Peer review of "Macroalgal metabolism and lateral carbon flows can create significant carbon sinks"

_Biogeosciences, 2019_

## Referee Comment (RC1) · Albert Pessarrodona Silvestre (Referee) · 10 Dec 2019

This is a great study that provides some highly valuable and relevant new insights about the potential transport of macroalgal carbon. Although the export of DOC below the mixed layer is believed to be the main pathway through which macroalgal carbon gets sequestered in the ocean, our understanding of the fate of macroalgal DOC after its release is very limited. This study presents tempting evidence of its potential export to offshore waters (but see some concerns below), which is an important step to verify the role of macroalgae in oceanic carbon sequestration. Overall, I found the study to be well conducted and well written. The authors provide a set of comprehensive measurements of different carbon compartments and forms, which I applaud. Although I am not familiar with some of the more technical protocols of the sample analysis,

further reading and consulting suggest that they are standard.

One of my principal concerns is that the authors have not yet established a direct transport link between the water exported from the macroalgal bed and the waters at the offshore site. The authors found that (1) water near the macroalgal bed had different properties (namely: lower DIC, fCO2 and higher DOC concentrations) than the water offshore — except for February, when DOC concentrations were not significantly different. They then used mass balance models to simulate the diurnal changes in the carbonate and DOC system of the macroalgal bed (ln. 148); incorporating water exchange into their models helped better explain their readings (ln. 218, 245), which suggests that (2) there is water inflowing and outflowing at both the macroalgal bed and offshore site. There is however no direct demonstration that it is specifically the macroalgal bed water the one that reaches the offshore waters. This is a very important nuance, as the water that lowers the CO2 concentrations and enhances atmospheric CO2 uptake at the offshore site could come from other habitats that "produce" low DIC, high DOC water (e.g. seagrass meadow). Characterizing the DOC profile of both waters could help shed light on the provenance of that water.

The mass balance models only consider changes due to processes related to macroalgal metabolism, but some could argue that they are missing some parameters. For example, volatile and semi-volatile compounds can be an important fraction of the DOC, and can be volatilize to the atmosphere (Ruiz-Halpern, Vaquer-Sunyer, & Duarte, 2014) instead of remaining in the water column as assumed here. Similarly, some of the other processes that can affect the DIC pools (e.g. dissolution, chemical addition; (Langdon et al., 2003)), are not considered. If the authors consider that those fluxes are negligible that is fine, but they should provide evidence to back their approach.

It is very valuable that the study measurements were conducted at two separate time points —albeit in the same season—which gives an idea of the variability associated with the carbon flows estimated in the study. For instance, both the amount of DOCM and its constituents (as suggested by the different decomposition rates) were different

across months (Wada et al., 2008). These points should be further elaborated to produce a rich and interesting discussion section. It would also be worth discussing how other species of macroalgae may differ in the production and characteristics of their DOC, as S. horneri was not the dominant species in the bed. Another limitation worth discussing is that DOC incubations for the degradation experiments were also maintained at a constant temperature (22), which may not necessarily reflect conditions in the field.

Finally, some of the sections of the manuscript also need to be further clarified, as it is difficult for the reader to grasp how some very key parameters where calculated. For example, it is unclear how the gross community production, respiration and calcification were calculated from the DOC bag experiments (ln. 160), all of which are key parameters in the model. It is also not very clear how the tidal water exchange (EXtide) rate was estimated from changes in depth (ln. 169)

Specific comments

Ln 33: Add "far" before "been"

Ln. 37 Add "more" before "efficiently"

Ln 45: stored where? In the sediments, water column. . .? Also, consider citing here (Queirós et al., 2019), which provides an example of macroalgal-sediment connectivity.

Ln. 52: I suggest making the topic sentence of the paragraph the fact that DOC is believed (at least according to (Krause-Jensen & Duarte, 2016)) to be the principal pathway of macroalgal carbon sequestration (although). This will highlight more the relevance of this study, as more empirical support is needed to demonstrate the assumptions of (Krause-Jensen & Duarte, 2016)

Ln. 55: This paragraph feels a bit out of place here, you are talking about DOC and all of a sudden start talking about the carbonate system. Consider rearranging/rewriting.

Ln. 61: The sentence gives the impression that the effects of macroalgal metabolism

in their surrounding waters have not been studied, which is not the case (the authors provide plenty of examples). What is truly novel is examining its effects on other water bodies. I suggest deleting "both macroalgal beds and"

Ln 67: Sargassaceous algae sounds a bit strange to me, perhaps just use Sargassum beds? Sargassums are also commonly found in tropical regions (Fulton et al., 2019), so I would suggest changing for "we focused on Sargassum beds because they are one of the dominant macroalgal habitats in temperate and tropical regions).

Ln. 69 The issue of carbon sequestration was not directly addressed in this paper, as no evidence that the carbon measured is locked away from the atmosphere for very long periods of time (decades-centuries) is presented. Although some of the DOC did not decompose after 150 days under constant experimental conditions, it is not known how long it would remain in the field or whether it could reach the mixed layer. I suggest cutting similar claims made throughout the ms

Ln. 75. Given that the water inflowing and outflowing from the bed is so important for this study, the readers would appreciate more details about the water movements around the study area (e.g. tidal characteristics, exposure)

Ln. 79. This sentence is a bit redundant from the one in Ln 76. Consider merging them.

Ln. 96. Is that the volume of seawater in the bag?

Ln. 109. Please indicate the pore size of the filter. Was the filtering pressurized?

Ln. 127. What concentration of KHPh?

Ln 140. At what height was the wind speed measured at Agenosho?

Ln 143. Delete "that"

Ln. 148. Using the active voice is more readable in this instance. "We simulated the diurnal changes and budgets of the carbonate system and DOC in the macroalgal bed

using mass balance models"

Ln. 151. This sentence seems to indicate that you changed the depth of the macroalgal bed. Please rewrite. Was the tide simulated by changing water height over the bed?

Ln 152. The average Sargassum biomass used was derived from the field surveys, right? Please state so

Ln 157. The amount of formulas, acronyms and parameters used in the manuscript can be a bit overwhelming. I encourage the authors to consider making a first figure with a schematic diagram of the different carbon pools and fluxes, as well as different carbon forms (e.g. POC, PIC, DOC, DIC. . .) and the processes that affect them (e.g. primary production, calcification. . .). That figure could include the formulas in lines 157-159 to show how they were calculated in the mass balance models. I think this could be very useful to the reader.

Ln. 160. It is very unclear how all these parameters where calculated. Did you use some sort of relationship between DOC release and productivity? Please provide further details.

Lns 165-166. They can be just one sentence

Lns 192-193. They can be just one sentence

Ln 205. The use of "g WW" is more standard. Also wet weight (WW) needs to be abbreviated somewhere in the paper.

Ln. 208-209. Please provide statistical evidence that the decrease in time is statistically significant.

Ln 210. Perhaps it would be informative to include those final percentages in Fig. 4, as the decrease is a bit hard to observe in some panels (e.g. 4b)

Ln. 218. Please provide an index of how well the model fits the data. This way you can say that a model improves or worsens by adding/removing water exchange.

Ln 238. Add "For example", before DIC uptake"

Ln. 168: The estimation of water exchange is crucial for the aims of this paper. I am having a bit of trouble understanding how you EXtide was estimated from changes in depth. Is that referring to tidal height? It could be helpful if some example values are provided (e.g. is the number greater on spring tides, what is the maximum value it can attain? 1? What would that mean)

Ln. 256. I wonder how seasonality will affect the fate of the DOC released as well. How do oceanographic conditions vary in the study area?

Ln 274. You may also be interested in the extensive work of Sophie Martin in maerl beds e.g. (Martin, Clavier, Chauvaud, & Thouzeau, 2007)

Ln. 296-297. These two statements seem contradictory

Ln. 306. Very interesting find!

Ln. 320. Insert "considered as" before "are"

Ln. 321. Consider "[. . .] export of particulate macroalgal carbon (e.g. entire thalli and fragments) to the deep sea [. . .]"

Figure 4. Consider stating the percentage of DOC remaining in each of the treatments of panels 4a and 4b as it is a big hard to tell how much remained sometimes. Also consider shading the area between the two treatments and indicating that it corresponds to the macroalgal DOC (DOCM; ln. 121).

Figure 5. I think that plotting the value of EX in this graphs would be very valuable, as it would help the reader understand what is the water doing (inflow or outflow), and how this affects the readings at the macroalgal bed and offshore sites. The mass balance model should also predict the observations at the offshore site; please plot those ones as well.

Figure 6. I suggest putting a dashed line through the middle of the panes to

[Figure]

clearly delineate the offshore waters from the macroalgal bed. Also, put the titles of "Offshore" and "Macroalgal bed" at the very top so it is easier to read. I think that using symbols instead of the photo of the macroalgal bed would declutter the figure and make it more understandable. For instance, the ones at https://ian.umces.edu/imagelibrary/displayimage-search-0-4529.html are freely available (with attribution) and make for very appealing figures.

References Fulton, C. J., Abesamis, R. A., Berkström, C., Depczynski, M., Graham, N. A. J., Holmes, T. H., . . . Wilson, S. K. (2019). Form and function of tropical macroalgal reefs in the Anthropocene. Functional Ecology, (January), 1–11. https://doi.org/10.1111/1365-2435.13282

Krause-Jensen, D., & Duarte, C. M. (2016). Substantial role of macroalgae in marine carbon sequestration. Nature Geoscience, 9(September), 737–742. https://doi.org/10.1038/ngeo2790

Langdon, C., Broecker, W. S., Hammond, D. E., Glenn, E., Fitzsimmons, K., Nelson, S. G., . . . Bonani, G. (2003). Effect of elevated CO 2 on the community metabolism of an experimental coral reef . Global Biogeochemical Cycles, 17(1). https://doi.org/10.1029/2002gb001941

Martin, S., Clavier, J., Chauvaud, L., & Thouzeau, G. (2007). Community metabolism in temperate maerl beds. I. Marine Ecology Progress Series, 335, 31–41. https://doi.org/10.3354/meps335031

Queirós, A. M., Stephens, N., Widdicombe, S., Tait, K., Mccoy, S. J., Ingels, J., . . . Somerfield, P. (2019). Connected macroalgal-sediment systems : blue carbon and food webs in the deep coastal ocean. Ecological Monographs, 0(0), e01366. https://doi.org/10.1002/ecm.1366

Ruiz-Halpern, S., Vaquer-Sunyer, R., & Duarte, C. M. (2014). Annual benthic metabolism and organic carbon fluxes in a semi-enclosed Mediterranean bay dominated by the macroalgae Caulerpa prolifera. Frontiers in Marine Science, 1(DEC), 1–10. https://doi.org/10.3389/fmars.2014.00067

Wada, S., Aoki, M. N., Mikami, A., Komatsu, T., Tsuchiya, Y., Sato, T., ... Hama, T. (2008). Bioavailability of macroalgal dissolved organic matter in seawater. Marine Ecology Progress Series, 370, 33–44. https://doi.org/10.3354/meps07645
* * *

---

## Referee Comment (RC2) · D. Krause-Jensen (Referee) · 30 Dec 2019

**GENERAL COMMENTS**

This study documents the exchange of dissolved carbon between a macroalgal habitat and adjacent waters. The study highlights that macroalgal metabolism and excretion of dissolved organic carbon (DOC) during the productive phase of the vegetation create low CO2 concentration and high DOC concentration that, via water exchange, propagates from the macroalgal habitat to waters beyond the habitat. The low CO2 concentrations created by macroalgae thereby contribute to increased air-sea CO2 uptake both at the habitat and beyond, and export of DOC beyond the habitat suggests a potential of this carbon to reach oceanic sinks. These findings add significantly

to the limited field evidence of the effect of macroalgal metabolism on dissolved carbon concentrations and the size of macroalgal-associated carbon fluxes and potentials for C-sequestration. Such evidence is important to underpin the recent notion that macroalgae contribute significantly to global C-sequestration, with the majority of the sequestration being supported by dissolved organic carbon reaching oceanic C-sink. The combination of in-situ measurements and flux studies, degradation experiments and modeling strengthen the findings. The study can be improved by adding detail in the method description, presentation and discussion of results and reference to earlier findings.

SPECIFIC COMMENTS

Methods

Field surveys (l. 76-80): - Please specify that the field studies were conducted during a diurnal cycle in February and March, respectively, and please underline the timing of sampling as well as the timing of sunrise/sunset so that the reader knows how the diurnal cycle was represented. Please also mention that February /March is the local winter time.

Field bag experiments: - (l. 93) Was the ambient seawater for the macroalgal bags and control bags taken from the macroalgal site?

Biomass, cover and species composition (l. 100-106): - How long were the transect lines? While cover was assessed every 10 m (in 1 x 1 m quadrates) it is not clear how biomass assessments relate to cover assessments. Were the five 0.5 x 0.5 m biomass samples taken within quadrates assessed for cover and where cover estimates documented dominance by sargassaceous algae? Or were the biomass samples placed e.g. randomly within the belt dominated by sargassaceous algae? Please add detail.

Degradation experiment - Why were the samples stored at room temperature (22 C)? Did this correspond to the in situ temperature? Were the samples aerated

or did they turn anoxic during the incubation? How were degradation rates calculated?

Mass balance modeling - It is not entirely clear how this modeling was conducted - please expand the explanation. As far as I understand, the modeling was conducted solely for the macroalgal site (and not for the off shore site), please state this clearly. - L. 154: Are the initial values for the macroalgal site estimated to be diurnal averages measured at the off shore site? (Please indicate in the text that the initial values are denotated "0" in the formula). (To clarify, I suggest moving the sentence (l. 163) "The values of DICO, TAlkO, and DOCO were the mean values at station H5." to follow l. 154.) L. 160-162: Please explain in more detail how the central parameters GCP, R and CC were determined (based on start/end and light/dark and macroalgal/control measurements) and which day length was applied. Regarding the calculation of calcification – please also see e.g. Wahl et al. 2018. - Please make it clear in the methods section how the two different model scenarios (i.e. with and without considering water exchange, blue line and black line of Fig 5) were calculated.

Results

(3.1) Carbonate system and DOC - Net community calcification: Did your study allow calculating potential differences in calcification between light and dark settings? - Fig. 2: Are the two lines significantly different?

(3.2) Biomass/cover - Fig. 3: Relationships between cover and biomass (relates to the question on how biomass samples were taken): How come that the highest biomass in panel a corresponds to the lowest coverage in panel b? And that the lowest biomass of sargassaceous algae (and highest relative biomass of "others" in panel c corresponds to a high (absolute and relative) cover of sargassaceous algae in panel d? Are there any significant biomass-cover relationships?

(3.3) Degradation of DOC - l. 211-212 "Degradation rates (k) estimated by exponential fitting were 0.0044 d−1 and 0.0018 d−1 in February and March, respectively." Please clarify how degradation rates were calculated by e.g. chnagingt he sentence to "Degradation rates (k) estimated by exponential fitting of XXX vs XXX were 0.0044 d−1 and 0.0018 d−1 in February and March, respectively." and specify XXX. Fig. 4: - It is notable that DOC concentrations of the control bags were similar between Feb and Mar while DOC concentrations of macroalgal bags differed considerably, with final concentrations being about 140 ïA■M in Feb and <100 ïA■M in Mar. Please discuss this in the discussion section.. - The 4th control sampling for March has high variability – might one sample be contaminated and should be omitted? - Panel c: Should the line-fit not be exponential? - How were degradation rates calculated. Based on fits of data in panel c? Why not based on fitting an exp decline curve to the macroalgal data of panel a and b?

(3.4) Carbon budgets - L. 214. "The mass balance models simulated the temporal changes in carbonate chemistry and DOC concentration (Fig. 5)." It is not clear how the mass balance models did this - please expand in the methods section and also elaborate a bit more here. Please explain that the model simulations represent both the situation when water exchange is taken into account (blue lines in Fig. 5) and the situation when it is not (black lines in Fig. 5). - I also suggest adding more detail to the legend of Fig. 5 to specify the significance of the blue line (in contrast to the black line), which is not mentioned in the current version of the legend. l. 216: How were the spans in hourly exchange rates calculated (35-48% and 50-76%) and why is the range not reported in Table 3 (35% and 50% is reported without any range). - It follows from the modeling approach (l. 223-224) that "DIC budgets driven by water exchange indicated a net input of DIC from offshore to the macroalgal bed (Fig. 6 and Table 3.)", - because otherwise the DIC levels at the macroalgal site would have been considerably lower that what was measured (as shown in Fig 5). However, the abstract says "The exported water lowered CO2 concentrations in the offsite surface water and enhanced atmospheric CO2 uptake.", and I think this statement needs be better underpinned by results and discussion.

Discussion

- Net calcification (NCC): Please discuss /mention what may constitute the NCC: calcareous algae in the algal bed, mussels. . . ? Did you identify any variation in NCC between light and dark incubations? – please mention/discuss. For these discussions I suggest referring/comparing to e.g. Wahl et al 2018 & Duarte & Krause-Jensen 2018.

- L. 276-8: Please elaborate a bit on this in relation to the differences observed between the Feb and Mar measurements. Regarding "growth phase", please mention that the study took place during the productive period.

- L. 305-307: Regarding the comparation between DOC turnover times: Were the experimental conditions similar?

- Please discuss reasons for the big difference in the residual concentrations of DOC in the degradation of material from Feb and March.

- C-sequestration. L. 322-323: I suggest mentioning that a first-order assessment suggested that almost 70% of global macroalgal C-sequestration is attributable to DOC export to the deep sea (Krause-Jensen & Duarte 2016).

Figs/Tables (in addition to what is mentioned above) Fig 5 legend: Please mention that details regarding rates are available in Table 1. I suggest moving Table 1 to supplementary material as the data are already presented in Fig 5.

Fig 6 & Legend: Please change "Carbon flows.." to "Dissolved carbon flows.." Please add explanation of RDOCm. Unclear what is the difference between NDR and DOC. The legend says that (107) and (88) represents DOC flows; then why are the numbers at the DOC arrows different? Please mention how the data were generated (model, degradation exp, bag exp – maybe using different colors). Mention that the same calcification rates are reported both as calcification rates and as inorganic biomass growth. Mention if the NCP is the sum of macroalgal and planktonic NCP.. Mention that details regarding rates of C-metabolism are available in Table 2 and details on water exchange rates are available in Table 3.

References

- It would be appropriate to mention the pioneer study by Smith 1981 suggesting that lowering of CO2 concentrations by macroalgae leads to increased CO2 uptake and to highlight that the current study not only confirms that this is the case but also takes the finding further by documenting that the effect extends beyond the macroalgal habitat.

- Line 60-61: "However, the effects of macroalgal metabolism on the carbonate system in both macroalgal beds and adjacent water bodies have not been quantified": Please consider rephrasing to say e.g. that there is limited field evidence on this.. Earlier studies have documented effects of macroalgal metabolism on the carbonate system (e.g. Wahl et al. 2018, Middelboe et al. 2007, Krause-Jensen et al. 2015 & 2016, Duarte & Krause-Jensen 2018), and some of these provide evidence of how diurnal/temporal variations in macroalgal metabolism affect calcification. You may also want to mention that there are recent studies on particulate organic carbon (POC) fluxes from macroalgae (e.g. Filbee-Dexter et al. 2018, Pedersen et al. 2019, Queirós et al. 2019...) but less information on DOC fluxes despite the expected major importance of macroalgal DOC fluxes for carbon sequestration.

Duarte, C. M., & Krause-Jensen, D. (2018). Greenland Tidal Pools as Hot Spots for Ecosystem Metabolism and Calcification. Estuaries and coasts, 41(5), 1314-1321.

Filbee-Dexter, K., Wernberg, T., Norderhaug, K. M., Ramirez-Llodra, E., & Pedersen, M. F. (2018). Movement of pulsed resource subsidies from kelp forests to deep fjords. Oecologia, 187(1), 291-304.

Krause-Jensen, D., Marbà, N., Sanz-Martin, M., Hendriks, I. E., Thyrring, J., Carstensen, J., ... & Duarte, C. M. (2016). Long photoperiods sustain high pH in Arctic kelp forests. Science advances, 2(12), e1501938.

Middelboe, A. L., & Hansen, P. J. (2007). High pH in shallow-water macroalgal habitats. Marine Ecology Progress Series, 338, 107-117.

Pedersen, M. F., Filbee-Dexter, K., Norderhaug, K. M., Fredriksen, S., Frisk, N. L., & Wernberg, T. (2019). Detrital carbon production and export in high latitude kelp forests. Oecologia, 1-13.

Smith, S. V. (1981). Marine macrophytes as a global carbon sink. Science, 211(4484), 838-840.

Queirós, A. M., Stephens, N., Widdicombe, S., Tait, K., McCoy, S. J., Ingels, J., ... & Cazenave, P. (2019). Connected macroalgal‐sediment systems: blue carbon and food webs in the deep coastal ocean. Ecological Monographs, e01366.

Wahl, M., Schneider Covachã, S., Saderne, V., Hiebenthal, C., Müller, J. D., Pansch, C., & Sawall, Y. (2018). Macroalgae may mitigate ocean acidification effects on mussel calcification by increasing pH and its fluctuations. Limnology and Oceanography, 63(1), 3-21.

TECHNICAL CORRECTIONS

l. 17-18: I suggest adding field measurements of carbon species to the set of applied methods. l. 22: Please change "offsite" to "offshore" l. 36: "is more labile" – I suggest rephrasing to underline the variable lability of macroalgal carbon. l. 39: "is comparable to.." – or larger than? l. 40-41: "Macroalgal beds therefore have the potential to sequester substantial amounts of carbon in marine systems": This does not follow logically from the previous sentences – please rephrase. l. 49-50: please mention also the estimated contribution of DOC to macroalgal C-sequestration in this previous study to highlight the hypothesis that macroalgal DOC is of major importance. l. 67: "..this macroalgae is the dominant group in temperate regions": should it be e.g.". . .the Sargassaceae family of macroalgae is a dominant group in temperate regions"? l. 74: "at a water depth.." –> "at water depths.." l. 188 and 193: Please add "(Table 1)" at the end of the sentence. l. 236: Please add after "(Table 2)" e.g.: ".., the rest being attributable to planktonic NCP." l. 241: please substitute "known" by "shown". l. 275/6: I suggest changing "The inhibition of macroalgal R by low water temperatures during the

winter can explain the relatively high NCP values during the productive period at our study site (Table 1 and 2)." to "The inhibition of macroalgal R by low water temperatures during the productive winter can explain the relatively high NCP values observed at our study site (Table 1 and 2)." - should "irradiation" be "irradiance"?

---

## Author Comment (AC1) · 21 Jan 2020

Author response to RC1 by Albert Pessarrodona Silvestre

We thank to your constructive comments. Below is reviewer's comments and our response to them.

Comment #1: This is a great study that provides some highly valuable and relevant new insights about the potential transport of macroalgal carbon. Although the export of DOC below the mixed layer is believed to be the main pathway through which macroalgal carbon gets sequestered in the ocean, our understanding of the fate of macroalgal DOC after its release is very limited. This study presents tempting evidence of its potential export to offshore waters (but see some concerns below), which is an important

step to verify the role of macroalgae in oceanic carbon sequestration. Overall, I found the study to be well conducted and well written. The authors provide a set of comprehensive measurements of different carbon compartments and forms, which I applaud. Although I am not familiar with some of the more technical protocols of the sample analysis, further reading and consulting suggest that they are standard.

Response: Thank you for your comments and suggestions. We have modified the manuscript considering your suggestions. Please see details below.

Change: We have modified the manuscript considering your suggestions. Please see details below.

Comment #2: One of my principal concerns is that the authors have not yet established a direct transport link between the water exported from the macroalgal bed and the waters at the offshore site. The authors found that (1) water near the macroalgal bed had different properties (namely: lower DIC, fCO2 and higher DOC concentrations) than the water offshore except for February, when DOC concentrations were not significantly different. They then used mass balance models to simulate the diurnal changes in the carbonate and DOC system of the macroalgal bed (ln. 148); incorporating water exchange into their models helped better explain their readings (ln. 218, 245), which suggests that (2) there is water inflowing and outflowing at both the macroalgal bed and offshore site. There is however no direct demonstration that it is specifically the macroalgal bed water the one that reaches the offshore waters. This is a very important nuance, as the water that lowers the CO2 concentrations and enhances atmospheric CO2 uptake at the offshore site could come from other habitats that "produce" low DIC, high DOC water (e.g. seagrass meadow). Characterizing the DOC profile of both waters could help shed light on the provenance of that water.

Response: We agree with your comment. Our results show that low CO2 and high DOC water is exported from the macroalgal bed but this finding does not directly demonstrate that the macroalgal bed water reaches the offshore waters and affects

carbon dynamics although we did not find such a low CO2 and high DOC water body mass around. Thus, we have rephrased the sentences about this claim throughout the manuscript. We have also added the discussion about the necessity of future study about the origin of water bodies affected by coastal vegetation.

Change: We have rephrased the sentences about this claim in Abstract, section 4.1, and section 5 as that macroalgal beds "potentially" create areas of adjacent water that serve as CO2 sinks. We have also added the discussion about the necessity of future study about the origin of water bodies affected by coastal vegetation.

Comment #3: The mass balance models only consider changes due to processes related to macroalgal metabolism, but some could argue that they are missing some parameters. For example, volatile and semi-volatile compounds can be an important fraction of the DOC, and can be volatilize to the atmosphere (Ruiz-Halpern, Vaquer-Sunyer, & Duarte, 2014) instead of remaining in the water column as assumed here. Similarly, some of the other processes that can affect the DIC pools (e.g. dissolution, chemical addition; (Langdon et al., 2003)), are not considered. If the authors consider that those fluxes are negligible that is fine, but they should provide evidence to back their approach.

Response: In this study, "DOC" did not contain volatile fraction of DOC because we measured DOC as non-purgeable organic carbon according to the well-established method. Because the model only calculated non-purgeable DOC, the volatilization of DOC can be ignored. About the DIC pools, carbonate dissolution and calcification were included in the mass balance model (Eq. 4).

Change: We have specified that our DOC was non-purgeable organic carbon.

Comment #4: It is very valuable that the study measurements were conducted at two separate time points albeit in the same season which gives an idea of the variability associated with the carbon flows estimated in the study. For instance, both the amount of DOCM and its constituents (as suggested by the different decomposition rates) were

different across months (Wada et al., 2008). These points should be further elaborated to produce a rich and interesting discussion section. It would also be worth discussing how other species of macroalgae may differ in the production and characteristics of their DOC, as S. horneri was not the dominant species in the bed. Another limitation worth discussing is that DOC incubations for the degradation experiments were also maintained at a constant temperature (22), which may not necessarily reflect conditions in the field.

Response: The difference in the initial DOCM concentrations of macroalgae bags between February and March would be caused by the differences in the biomass and water volume in the experimental bags. We have added the discussion about this point. We have added the discussion about seasonal and interspecific variations in the release rates of refractory DOC by referring previous works. In this study, we used room temperature (22°C), which is higher than in situ temperature, for both study periods to compare the quality of organic matter. We have added the discussion about the effect of temperature on the microbial degradation of DOC in the discussion section.

Change: We have added the discussion about this point as follows: "The difference in the initial DOCM concentrations of macroalgae bags between February and March would be caused by the differences in the biomass and water volume in the experimental bags (Fig. 4a, b). The variation of DOC concentration may affect the degradation rates via resource limitations for microbial activity (e.g., Arrieta et al., 2015). Understanding the fate of macroalgal DOC will be supported by assessing physical and biochemical factors regulating the microbial degradation of DOC." We have added the discussion about seasonal and interspecific variations in the release rates of refractory DOC as follows: "Phlorotannin contents in macroalgal thalli have variations among seasons, growth phases, and species (Steinberg, 1989; Kamiya et al., 2010), which may regulate seasonal and interspecific variations in the biological recalcitrance of macroalgal DOC." In this study, we used room temperature (22°C), which is higher than in situ temperature for both study periods to compare the quality of organic matter. We have added the discussion about the effect of temperature on the microbial degradation of DOC in the discussion section.

Comment #5: Finally, some of the sections of the manuscript also need to be further clarified, as it is difficult for the reader to grasp how some very key parameters where calculated. For example, it is unclear how the gross community production, respiration and calcification were calculated from the DOC bag experiments (ln. 160), all of which are key parameters in the model. It is also not very clear how the tidal water exchange (EXtide) rate was estimated from changes in depth (ln. 169)

Response: We agree with your comment.

Change: We have added the equations and explanations for metabolic parameter estimation in Materials and methods section. We have added the equation for calculating EXtide for clarification and temporal changes in EX with water height in Fig. 5.

Specific comments Comment #6: Ln 33: Add "far" before "been"

Response: We agree with your comment.

Change: We have added "far" before "been" as per your suggestion.

Comment #7: Ln. 37 Add "more" before "efficiently"

Response: We agree with your comment.

Change: We have added "more" before "efficiently" as per your suggestion.

Comment #8: Ln 45: stored where? In the sediments, water column...? Also, consider citing here (Queirós et al., 2019), which provides an example of macroalgal-sediment connectivity.

Response: We agree with your comment.

Change: We have added "stored in sediments and water column" and the citation "Queirós et al., 2019" as per your suggestion.

Comment #9: Ln. 52: I suggest making the topic sentence of the paragraph the fact that DOC is believed (at least according to (Krause-Jensen & Duarte, 2016)) to be the principal pathway of macroalgal carbon sequestration (although). This will highlight more the relevance of this study, as more empirical support is needed to demonstrate the assumptions of (Krause-Jensen & Duarte, 2016)

Response: We agree with your comment.

Change: We have made the topic sentence as follows: "The export and persistence of macroalgal dissolved organic carbon (DOC) is proposed to be one of the principal pathways of macroalgal carbon sequestration but more empirical support is needed to quantify this carbon flow."

Comment #10: Ln. 55: This paragraph feels a bit out of place here, you are talking about DOC and all of a sudden start talking about the carbonate system. Consider rearranging/rewriting.

Response: We agree with your comment.

Change: We have added the following sentence at the top of this sentence: "Even though macroalgal beds have the significant function of assimilating organic carbon, they could also be net $CO_2$ emitters via air–water $CO_2$ exchange depending on the chemical equilibrium in carbonate system in water columns."

Comment #11: Ln. 61: The sentence gives the impression that the effects of macroalgal metabolism in their surrounding waters have not been studied, which is not the case (the authors provide plenty of examples). What is truly novel is examining its effects on other water bodies. I suggest deleting "both macroalgal beds and"

Response: We agree with your comment.

Change: We have deleted this phrase as per your suggestion.

Comment #12: Ln 67: Sargassaceous algae sounds a bit strange to me, perhaps

just use Sargassum beds? Sargassums are also commonly found in tropical regions (Fulton et al., 2019), so I would suggest changing for "we focused on Sargassum beds because they are one of the dominant macroalgal habitats in temperate and tropical regions).

Response: We agree with your comment.

Change: We have replaced "Sargassaceous" to "Sargassum" through the manuscript. Also, we have changed the sentence as per your suggestion.

Comment #13: Ln. 69 The issue of carbon sequestration was not directly addressed in this paper, as no evidence that the carbon measured is locked away from the atmosphere for very long periods of time (decades-centuries) is presented. Although some of the DOC did not decompose after 150 days under constant experimental conditions, it is not known how long it would remain in the field or whether it could reach the mixed layer. I suggest cutting similar claims made throughout the ms

Response: The elucidation of the mechanisms of long-term $CO_2$ sequestration by macroalgae is a final goal of our study but the present study addressed the mechanism of $CO_2$ uptake by macroalgae. We have modified this point through the manuscript as per your suggestion.

Change: We have clarified our research goal as the assessment of the key mechanisms of $CO_2$ uptake by the macroalgal bed in Abstract and Introduction section. (See also comment #2)

Comment #14: Ln. 75. Given that the water inflowing and outflowing from the bed is so important for this study, the readers would appreciate more details about the water movements around the study area (e.g. tidal characteristics, exposure)

Response: We agree with your comment.

Change: We have added the following sentence as per your suggestion: "The study site is characterized by relatively high tidal amplitude (<4 m) and adjacent to a deep

strait (∼60 m).”

Comment #15: Ln. 79. This sentence is a bit redundant from the one in Ln 76. Consider merging them.

Response: We agree with your comment.

Change: We have modified this sentence as follows: “Surface water samples for analyses of DIC, TAlk, and DOC were collected from a research vessel three times during the daytime in both survey at five stations (H1–H5).”

Comment #16: Ln. 96. Is that the volume of seawater in the bag?

Response: Yes, this is the volume of seawater in each bag.

Change: We have added the sentence “in each bag” here.

Comment #17: Ln. 109. Please indicate the pore size of the filter. Was the filtering pressurized?

Response: We agree with your comment.

Change: We have added the information of pore size (0.7 $\mu$m) and filtering process (“under reduced pressure”).

Comment #18: Ln. 127. What concentration of KHPh?

Response: We have added the concentration of Potassium hydrogen phthalate (83, 166, and 332 $\mu$M).

Change: We have added the concentration of Potassium hydrogen phthalate.

Comment #19: Ln 140. At what height was the wind speed measured at Agenosho?

Response: The altitude was 6.5 m.

Change: We have added this information.

Comment #20: Ln 143. Delete "that"

Response: We agree with your comment.

Change: We have deleted "that" as per your suggestion.

Comment #21: Ln. 148. Using the active voice is more readable in this instance. "We simulated the diurnal changes and budgets of the carbonate system and DOC in the macroalgal bed using mass balance models"

Response: We agree with your comment.

Change: We have modified the sentence as per your suggestion.

Comment #22: Ln. 151. This sentence seems to indicate that you changed the depth of the macroalgal bed. Please rewrite. Was the tide simulated by changing water height over the bed?

Response: We simulated the tide by changing water height over the bed.

Change: For clarification, we have modified the sentence as follows: "..., and the tide was simulated by changing water height along with the observed tide."

Comment #22: Ln 152. The average Sargassum biomass used was derived from the field surveys, right? Please state so

Response: Yes, it is right.

Change: We have added the sentence "the average biomass of Sargassum algae obtained from the field survey".

Comment #23: Ln 157. The amount of formulas, acronyms and parameters used in the manuscript can be a bit overwhelming. I encourage the authors to consider making a first figure with a schematic diagram of the different carbon pools and fluxes, as well as different carbon forms (e.g. POC, PIC, DOC, DIC...) and the processes that affect them (e.g. primary production, calcification...). That figure could include the formulas

in lines 157-159 to show how they were calculated in the mass balance models. I think this could be very useful to the reader.

Response: We agree with your comment.

Change: We have added the new figure with a schematic diagram of the different carbon pools, fluxes and processes (Fig. 2 in the revised manuscript).

Comment #24: Ln. 160. It is very unclear how all these parameters where calculated. Did you use some sort of relationship between DOC release and productivity? Please provide further details.

Response: We agree with your comment.

Change: We have added the equations and explanations for metabolic parameter estimation in Materials and methods section.

Comment #25: Lns 165-166. They can be just one sentence

Response: We agree with your comment.

Change: We have modified the sentences as per your suggestion.

Comment #26: Lns 192-193. They can be just one sentence

Response: We agree with your comment.

Change: We have modified the sentences as per your suggestion.

Comment #27: Ln 205. The use of "g WW" is more standard. Also wet weight (WW) needs to be abbreviated somewhere in the paper.

Response: We agree with your comment.

Change: We have replaced "g-ww" to "g WW" through the manuscript and added the abbreviation in Materials and methods.

Comment #28: Ln. 208-209. Please provide statistical evidence that the decrease in

time is statistically significant.

Response: We agree with your comment. We have conducted a Welch's two-sample t-test to detect the differences between the initial and final concentrations of DOC during degradation experiments. The conclusion has not been changed.

Change: We have conducted a Welch's two-sample t-test to detect the differences between the initial and final concentrations of DOC during degradation experiments. We have added the analytical process in the Materials and methods section and the results of this analysis in the Results section.

Comment #29: Ln 210. Perhaps it would be informative to include those final percentages in Fig. 4, as the decrease is a bit hard to observe in some panels (e.g. 4b)

Response: We agree with your comment.

Change: We have added these final percentages in Fig. 4.

Comment #30: Ln. 218. Please provide an index of how well the model fits the data. This way you can say that a model improves or worsens by adding/removing water exchange.

Response: We have added the explanation for the model improvement (the change in the RMSEs of every parameters) by considering water exchange in this paragraph and the legend of Fig. 5. In the previous version of our manuscript, model fitting was performed by minimizing RMSEs solely for DIC model but it may cause the uncertainty in other parameters (i.e., TAlk, DOC, and fCO2). We have modified this model fitting method as follows: "EXr was determined by fitting the models so as to minimize the root mean squared error (RMSE) compared with the observed values. RMSEs were calculated for the z-scores of DIC, TAlk, DOC, and fCO2 values, which were standardized anomalies from the mean observed values divided by the standard deviations. The EXr value that minimize the averaged RMSEs for these four parameters was determined for each survey." This modification has changed the results of water exchange rate and

carbon budgets but the conclusion has not been changed.

Change: We have added the explanation for the model improvement (the change in the RMSEs of every parameters) by considering water exchange in this paragraph and the legend of Fig. 5. We have modified the model fitting method and the related results (Fig. 5 and Table 3).

Comment #31: Ln 238. Add "For example", before DIC uptake"

Response: We agree with your comment.

Change: We have added "For example," before DIC uptake" as per your suggestion.

Comment #32: Ln. 168: The estimation of water exchange is crucial for the aims of this paper. I am having a bit of trouble understanding how you EXtide was estimated from changes in depth. Is that referring to tidal height? It could be helpful if some example values are provided (e.g. is the number greater on spring tides, what is the maximum value it can attain? 1? What would that mean)

Response: We agree with your comment.

Change: We have added the equation for calculating EXtide for clarification and temporal changes in EX with water height in Fig. 5.

Comment #33: Ln. 256. I wonder how seasonality will affect the fate of the DOC released as well. How do oceanographic conditions vary in the study area?

Response: In this study, we did not collect the seasonal data for macroalgal DOC and oceanographic conditions, but we have added the discussion about this point (temperature and oceanographic conditions).

Change: We have added the discussion about the fate of macroalgal DOC depending on temperature and oceanographic conditions.

Comment #34: Ln 274. You may also be interested in the extensive work of Sophie

Martin in maerl beds e.g. (Martin, Clavier, Chauvaud, & Thouzeau, 2007)

Response: We agree with your comment.

Change: We have added this citation and its NCP value in the revised manuscript.

Comment #35: Ln. 296-297. These two statements seem contradictory

Response: We agree with your comment.

Change: For better clarification, we have modified the sentence as follows: "Although the DOC release rates were similar between our two surveys, the percentages were very different between February (34 %) and March (6 %) (Fig. 6)."

Comment #36: Ln. 306. Very interesting find!

Response: Thank you!

Change: We have added the discussion about seasonal and interspecific variations in the release rates of refractory DOC by referring previous work. (See also comment #4)

Comment #37: Ln. 320. Insert "considered as" before "are"

Response: We agree with your comment.

Change: We have modified this sentence as per your suggestion.

Comment #38: Ln. 321. Consider "[...] export of particulate macroalgal carbon (e.g. entire thalli and fragments) to the deep sea [...]"

Response: We agree with your comment.

Change: We have modified this sentence as per your suggestion.

Comment #39: Figure 4. Consider stating the percentage of DOC remaining in each of the treatments of panels 4a and 4b as it is a big hard to tell how much remained sometimes. Also consider shading the area between the two treatments and indicating that it corresponds to the macroalgal DOC (DOCM; ln. 121).

Response: We agree with your comment.

Change: We have added the final percentage values of DOC in each treatment inside the panels in Fig. 4. We have also shading the area between the two treatments and indicating that it corresponds to the DOCM.

Comment #40: Figure 5. I think that plotting the value of EX in this graphs would be very valuable, as it would help the reader understand what is the water doing (inflow or outflow), and how this affects the readings at the macroalgal bed and offshore sites. The mass balance model should also predict the observations at the offshore site; please plot those ones as well.

Response: We agree with your comment about EX. The mass balance model cannot predict the observations at the offshore site because the values of the offshore site were used as endmember of inflowing water.

Change: We have added the plot of EX along with water height in Fig. 5. We have added the explanation for clarifying that the values of the offshore site were used as endmember of inflowing water in the Materials and methods section.

Comment #41: Figure 6. I suggest putting a dashed line through the middle of the panes to clearly delineate the offshore waters from the macroalgal bed. Also, put the titles of "Offshore" and "Macroalgal bed" at the very top so it is easier to read. I think that using symbols instead of the photo of the macroalgal bed would declutter the figure and make it more understandable. For instance, the ones at https://ian.umces.edu/imagelibrary/displayimage-search-0-4529.html are freely available (with attribution) and make for very appealing figures.

Response: We agree with your comment.

Change: We have modified figure 6 as per your suggestion. We have put dashed lines for delineating the offshore waters from the macroalgal bed. We have also put the titles of "Offshore" and "Macroalgal bed" at the top. We have made symbols of macroalgae

and used them.

---

## Author Comment (AC2) · 21 Jan 2020

Author response to RC2 by Dorte Krause-Jensen

We thank to your constructive comments. Below is reviewer's comments and our response to them.

GENERAL COMMENTS Comment #1: This study documents the exchange of dissolved carbon between a macroalgal habitat and adjacent waters. The study highlights that macroalgal metabolism and excretion of dissolved organic carbon (DOC) during the productive phase of the vegetation create low CO2 concentration and high DOC concentration that, via water exchange, propagates from the macroalgal habitat to waters beyond the habitat. The low CO2 concentrations created by macroalgae thereby

contribute to increased air-sea CO2 uptake both at the habitat and beyond, and export of DOC beyond the habitat suggests a potential of this carbon to reach oceanic sinks. These findings add significantly to the limited field evidence of the effect of macroalgal metabolism on dissolved carbon concentrations and the size of macroalgal-associated carbon fluxes and potentials for C-sequestration. Such evidence is important to underpin the recent notion that macroalgae contribute significantly to global C-sequestration, with the majority of the sequestration being supported by dissolved organic carbon reaching oceanic C-sink. The combination of in-situ measurements and flux studies, degradation experiments and modeling strengthen the findings. The study can be improved by adding detail in the method description, presentation and discussion of results and reference to earlier findings.

Response: Thank you for your comments and suggestions. We have modified the manuscript considering your suggestions. Please see details below.

Change: We have modified the manuscript considering your suggestions. Please see details below.

SPECIFIC COMMENTS Methods Comment #2: Field surveys (l. 76-80): - Please specify that the field studies were conducted during a diurnal cycle in February and March, respectively, and please underline the timing of sampling as well as the timing of sunrise/sunset so that the reader knows how the diurnal cycle was represented. Please also mention that February /March is the local winter time.

Response: Field surveys were conducted only during the daytime. For estimating diurnal cycles including nighttime, we used a field-bag method (especially respiration rate) and mass balance modelling (L154). We collected water samples three times at 10:00, 13:00, and 16:00 during the daytime (approximately from 7:00 to 17:00).

Change: We have added this information in this paragraph. We have also showed that February and March are included in the local winter period in this paragraph.

Comment #3: Field bag experiments: - (l. 93) Was the ambient seawater for the macroalgal bags and control bags taken from the macroalgal site?

Response: Yes, the ambient seawater for every treatment was collected in the macroalgal bed.

Change: We have added this information in the paragraph.

Comment #4: Biomass, cover and species composition (l. 100-106): - How long were the transect lines? While cover was assessed every 10 m (in 1 x 1 m quadrates) it is not clear how biomass assessments relate to cover assessments. Were the five 0.5 x 0.5 m biomass samples taken within quadrates assessed for cover and where cover estimates documented dominance by sargassaceous algae? Or were the biomass samples placed e.g. randomly within the belt dominated by sargassaceous algae? Please add detail.

Response: Both transect lines were 120 m. Five quadrats (0.5 m $\times$ 0.5 m) for quantifying biomass were randomly located in the area dominated by Sargassum algae along each transect.

Change: We have added this information in this paragraph.

Comment #5: Degradation experiment - Why were the samples stored at room temperature (22°C)? Did this correspond to the in situ temperature? Were the samples aerated or did they turn anoxic during the incubation? How were degradation rates calculated?

Response: In this study, we used room temperature (22°C), which is higher than in situ temperature, for both study periods to compare the quality of organic matter. We made the headspace to keep sufficient oxygen for aerobic degradation.

Change: We have added the discussion about the effect of temperature on the microbial degradation of DOC in the discussion section. We have added the explanation about making headspace in this paragraph. We have added the equation for calculat-
ing degradation rates (k).

Comment #6: Mass balance modeling - It is not entirely clear how this modeling was conducted -please expand the explanation. As far as I understand, the modeling was conducted solely for the macroalgal site (and not for the off shore site), please state this clearly. –

Response: As you say, this modeling was conducted solely for the macroalgal site and the observed values of offshore site were used as the endmembers of inflowing carbon to the macroalgal bed.

Change: We added the sentences to clarify this point in this paragraph.

Comment #7: L. 154: Are the initial values for the macroalgal site estimated to be diurnal averages measured at the off shore site? (Please indicate in the text that the initial values are denotated "0" in the formula). (To clarify, I suggest moving the sentence (l. 163) "The values of DICO, TAlkO, and DOCO were the mean values at station H5." to follow l.154.)

Response: We agree with your suggestion.

Change: We have moved this sentence and rephrased as follows: "The values of DICO, TAlkO, and DOCO were the mean values at the offshore station (H5). The initial values in the simulation were defined as the average values at the offshore site (station H5). Namely, DIC(0), Talk(0), and DOC(0) were DICO, TAlkO, and DOCO, respectively."

Comment #8: L. 160-162: Please explain in more detail how the central parameters GCP, R and CC were determined (based on start/end and light/dark and macroalgal/control measurements) and which day length was applied. Regarding the calculation of calcification – please also see e.g. Wahl et al. 2018.

Response: We agree with your suggestion. We used the day length shown in Table 2.

Change: We have added the equations and explanations for metabolic parameter estimation.

Comment #9: - Please make it clear in the methods section how the two different model scenarios (i.e. with and without considering water exchange, blue line and black line of Fig 5) were calculated.

Response: We agree with your suggestion.

Change: We have added the explanation of two different model scenarios in this paragraph.

Results Comment #10: (3.1) Carbonate system and DOC - Net community calcification: Did your study allow calculating potential differences in calcification between light and dark settings? - Fig. 2: Are the two lines significantly different?

Response: Because we used both transparent and dark bags for measuring net community calcification (NCC), we can calculate NCC rates in both light and dark settings (Table S3 in the Supplement). As we have discussed in the previous manuscript, it is difficult to discuss the differences in NCC values between light and dark settings because the uncertainty of NCC derived from the measurement precision were comparable to the observed values. We did not conduct statistical analysis here because we did not intend to discuss the differences between March and February.

Change: We have not changed manuscript about this comment.

Comment #11: (3.2) Biomass/cover - Fig. 3: Relationships between cover and biomass (relates to the question on how biomass samples were taken): How come that the highest biomass in panel a corresponds to the lowest coverage in panel b? And that the lowest biomass of sargassaceous algae (and highest relative biomass of "others" in panel c corresponds to a high (absolute and relative) cover of sargassaceous algae in panel d? Are there any significant biomass-cover relationships?

Response: As we replied to the question on how biomass samples were taken, quadrats for biomass and coverage were randomly located along each transect. We,

thus, think that the heterogenous colonization of macroalgae (e.g., patches) caused the inconsistence. Because we used only the averaged biomass for the mass balance modelling, this inconsistence between biomass and coverage did not change the conclusion.

Change: We have added the explanation about how biomass samples were taken in Materials and methods section.

Comment #12: (3.3) Degradation of DOC - l. 211-212 "Degradation rates (k) estimated by exponential fitting were 0.0044 d$-1$ and 0.0018 d$-1$ in February and March, respectively." Please clarify how degradation rates were calculated by e.g. chnagingt he sentence to "Degra-dation rates (k) estimated by exponential fitting of XXX vs XXX were 0.0044 d$-1$ and 0.0018 d$-1$ in February and March, respectively." and specify XXX.

Response: Thank you for your comment. We have added the explanation.

Change: We have added the explanation and equation for calculating Degradation rates (k) in Materials and methods section.

Comment #13: Fig. 4: - It is notable that DOC concentrations of the control bags were similar between Feb and Mar while DOC concentrations of macroalgal bags differed considerably, with final concentrations being about 140 $\mu$M in Feb and <100 $\mu$M in Mar. Please discuss this in the discussion section..

Response: The difference in the initial DOCM concentrations of macroalgae bags between February and March would be caused by the differences in the biomass and water volume in the experimental bags. We have added the discussion about this point.

Change: We have added the discussion about this point as follows: "The difference in the initial DOCM concentrations of macroalgae bags between February and March would be caused by the differences in the biomass and water volume in the experimental bags (Fig. 4a, b). The variation of DOC concentration may affect the degradation rates via resource limitations for microbial activity (e.g., Arrieta et al., 2015). The understanding of the fate of macroalgal DOC will be supported by assessing physical and biochemical factors regulating the microbial degradation of DOC."

Comment #14: - The 4th control sampling for March has high variability – might one sample be contaminated and should be omitted? - Panel c: Should the line-fit not be exponential?

Response: Thank you very much for your comment. We conducted triplicate analyses per one sample for DOC measurement using TOC analyzer to reduce the analytical uncertainty and used the average of this triplicate. One of the samples for 30-day control contained an erroneous value within the triplicate, which caused this unintentional high variability. We have omitted this erroneous value and modified Fig. 4b, c and related sentences. We rechecked every data and the others were not erroneous. After this correction, plots became suitable for exponential fitting ($R^2$ was improved).

Change: We have omitted the erroneous value and modified Fig. 4b, c and related sentences.

Comment #15: - How were degradation rates calculated. Based on fits of data in panel c? Why not based on fitting an exp decline curve to the macroalgal data of panel a and b?

Response: We have added the explanation and equation for calculating Degradation rates (k) in Materials and methods section. The focus on this degradation experiment was the quantification of refractory DOC derived from macroalgae. We decided not to fit exponential decay curve to the data of macroalgal bags in Fig. 4a, b because DOC of macroalgal bags contained ambient DOC.

Change: We have added the explanation and equation for calculating Degradation rates (k) in Materials and methods section.

Comment #16: (3.4) Carbon budgets - L. 214. "The mass balance models simulated the temporal changes in carbonate chemistry and DOC concentration (Fig. 5)." It is not clear how the mass balance models did this - please expand in the methods section and also elaborate a bit more here. Please explain that the model simulations represent both the situation when water exchange is taken into account (blue lines in Fig. 5) and the situation when it is not (black lines in Fig. 5). - I also suggest adding more detail to the legend of Fig. 5 to specify the significance of the blue line (in contrast to the black line), which is not mentioned in the current version of the legend.

Response: We have added the explanation of two different model scenarios in Materials and methods and result section. We have added the explanation for the model improvement (the change in the RMSEs of every parameters) by considering water exchange in this paragraph and the legend of Fig. 5. In the previous version of our manuscript, model fitting was performed by minimizing RMSEs solely for DIC model but it may cause the uncertainty in other parameters (i.e., TAlk, DOC, and fCO2). We have modified this model fitting method as follows: "EXr was determined by fitting the models so as to minimize the root mean squared error (RMSE) compared with the observed values. RMSEs were calculated for the z-scores of DIC, TAlk, DOC, and fCO2 values, which were standardized anomalies from the mean observed values divided by the standard deviations. The EXr value that minimize the averaged RMSEs for these four parameters was determined for each survey." This modification has changed the results of water exchange rate and carbon budgets but the conclusion has not been changed.

Change: We have added the explanation of two different model scenarios in Materials and methods and result section. We have added the explanation for the model improvement (the change in the RMSEs of every parameters) by considering water exchange in this paragraph and the legend of Fig. 5. We have modified the model fitting method and the related results (Fig. 5 and Table 3).

Comment #17: l. 216: How were the spans in hourly exchange rates calculated (35-

48% and 50-76%) and why is the range not reported in Table 3 (35% and 50% is reported without any range).

Response: EXr values were constants estimated for each survey but EXtide values were variables changing depending on water height. For clarification, we have rephrased this sentence as follows: "Hourly water exchange rates (the sum of EXtide and EXr) were...". We have showed the temporal change in water exchange rate (the sum of EXtide and EXr) in Fig. 5.

Change: We have rephrased this sentence as follows: "Hourly water exchange rates (the sum of EXtide and EXr) were...". We have showed the temporal change in water exchange rate (the sum of EXtide and EXr) in Fig. 5.

Comment #18: - It follows from the modeling approach (l. 223-224) that "DIC budgets driven by water exchange indicated a net input of DIC from offshore to the macroalgal bed (Fig. 6 and Table 3.)", - because otherwise the DIC levels at the macroalgal site would have been considerably lower that what was measured (as shown in Fig 5). However, the abstract says "The exported water lowered CO2 concentrations in the offsite surface water and enhanced atmospheric CO2 uptake.", and I think this statement needs be better underpinned by results and discussion.

Response: Thank you for your suggestion. We agree that our study did not directly demonstrate the enhancement of CO2 uptake in offshore site by the macroalgal bed.

Change: We rephrased the sentence in Abstract as follows: "These results indicate that the exported water would potentially lower CO2 concentrations in the offshore surface water and enhance atmospheric CO2 uptake."

Discussion Comment #19: - Net calcification (NCC): Please discuss /mention what may constitute the NCC: calcareous algae in the algal bed, mussels. . . ? Did you identify any variation in NCC between light and dark incubations? – please mention/discuss. For these discussions I suggest referring/comparing to e.g. Wahl et al 2018 & Duarte

& Krause-Jensen 2018.

Response: As described above, we can calculate NCC rates in both light and dark settings (Table S3 in the Supplement). However, it is difficult to discuss quantitatively the differences in NCC values between light and dark settings because the uncertainty of NCC derived from the measurement precision were comparable to the observed values. We also think that the quantitative comparison between our data and a previous work is difficult. However, we have cited this previous work about the calcification in macroalgal beds.

Change: We have cited this previous work about the calcification in macroalgal beds.

Comment #20: - L. 276-8: Please elaborate a bit on this in relation to the differences observed between the Feb and Mar measurements. Regarding "growth phase", please mention that the study took place during the productive period.

Response: In the present study, both surveys were conducted during the productive period but the averaged biomass per individual S. horneri used for field bag experiment was different (February, 353 g WW; March 260 g WW), which may indicate the difference in growth phase.

Change: We have added this explanation in this paragraph.

Comment #21: - L. 305-307: Regarding the comparation between DOC turnover times: Were the experimental conditions similar?

Response: Wada et al. (2008) calculated degradation rates for 30 days incubation, which was shorter than our study. Thus, we recalculated degradation rates for 30 days incubation and compared with Wada et al. (2008). This recalculation results also showed the same trend.

Change: We have modified this sentence according to this recalculation. We have also added the sentence regarding the temporal change in degradation rate during the DOC degradation referring a previous study.

Comment #22: - Please discuss reasons for the big difference in the residual concentrations of DOC in the degradation of material from Feb and March.

Response: As described above, the difference in the residual DOCM concentrations of macroalgae bags between February and March would be caused by the differences in the initial DOCM concentrations.

Change: We have added the discussion about this point as follows: "The difference in the initial DOCM concentrations of macroalgae bags between February and March would be caused by the differences in the biomass and water volume in the experimental bags (Fig. 4a, b). The variation of DOC concentration may affect the degradation rates via resource limitations for microbial activity (e.g., Arrieta et al., 2015). The understanding of the fate of macroalgal DOC will be supported by assessing physical and biochemical factors regulating the microbial degradation of DOC."

Comment #23: - C-sequestration. L. 322-323: I suggest mentioning that a first-order assessment suggested that almost 70% of global macroalgal C-sequestration is attributable to DOC export to the deep sea (Krause-Jensen & Duarte 2016).

Response: We agree with your suggestion.

Change: We have added this mentioning in this paragraph as per your suggestion.

Comment #24: Figs/Tables (in addition to what is mentioned above) Fig 5 legend: Please mention that details regarding rates are available in Table 1. I suggest moving Table 1 to supplementary material as the data are already presented in Fig 5.

Response: We showed Table 1 separately from Fig. 5 for clearly representing the difference in parameters between the macroalgal bed and the offshore by using statistical analysis. We believe that Table 1 should be also represented in main manuscript.

Change: We have not changed our manuscript.

Comment #25: Fig 6 & Legend: Please change "Carbon flows.." to "Dissolved carbon

flows..." Please add explanation of RDOCm. Unclear what is the difference between NDR and DOC. The legend says that (107) and (88) represents DOC flows; then why are the numbers at the DOC arrows different? Please mention how the data were generated (model, degradation exp, bag exp – maybe using different colors). Mention that the same calcification rates are reported both as calcification rates and as inorganic biomass growth. Mention if the NCP is the sum of macroalgal and planktonic NCP.. Mention that details regarding rates of C-metabolism are available in Table 2 and details on water exchange rates are available in Table 3.

Response: Because we also showed air–water CO2 gas exchange flux in this figure, we think that "Carbon flows..." is better explanation. We agree with the other comments.

Change: We have modified the legend as follows: "Carbon flows and community metabolism (NCP, net community production; NCC, net community calcification; NDR, net DOC release) in the macroalgal bed. NCP, NCC, and NDR were calculated using the results of field bag experiments (details are available in Table 2). Biomass growth in terms of organic carbon (OC) was calculated by subtracting NDR from NCP. Biomass growth in terms of inorganic carbon (IC) was same as NCC. DIC and DOC flows via water exchange were estimated by mass balance modelling (details are available in Table 3). Community metabolism, biomass growth, and DOC outflow indicates the sum of macroalgal and planktonic carbon flows. The parentheses show the carbon flows solely due to macroalgae. Carbon fluxes were calculated in units of mmoles per square meter of the surface area of the macroalgal bed per day. RDOCM indicates refractory DOC released by macroalgae."

References Comment #26: - It would be appropriate to mention the pioneer study by Smith 1981 suggesting that lowering of CO2 concentrations by macroalgae leads to increased CO2 uptake and to highlight that the current study not only confirms that this is the case but also takes the finding further by documenting that the effect extends beyond the macroalgal habitat.

Response: We agree with your comment.

Change: We have added the explanation about the previous study by Smith (1981).

Comment #27: - Line 60-61: "However, the effects of macroalgal metabolism on the carbonate system in both macroalgal beds and adjacent water bodies have not been quantified": Please consider rephrasing to say e.g. that there is limited field evidence on this.. Earlier studies have documented effects of macroalgal metabolism on the carbonate system (e.g. Wahl et al. 2018, Middelboe et al. 2007, Krause-Jensen et al. 2015 & 2016, Duarte & Krause-Jensen 2018), and some of these provide evidence of how diurnal/temporal variations in macroalgal metabolism affect calcification. You may also want to mention that there are recent studies on particulate organic carbon (POC) fluxes from macroalgae (e.g. Filbee-Dexter et al. 2018, Pedersen et al. 2019, Queirós et al. 2019...) but less information on DOC fluxes despite the expected major importance of macroalgal DOC fluxes for carbon sequestration.

Response: We agree with your comment.

Change: We have rephrased this sentence as follows: "Indeed, some previous studies have shown that macroalgal beds act as sinks for atmospheric $CO_2$ (Delille et al., 2009; Ikawa and Oechel, 2015; Koweek et al., 2017) and contribute to global carbon fluxes (Smith, 1981; Krause-Jensen and Duarte, 2016). Macroalgal metabolism regulates diurnal and temporal variations in carbonate chemistry and affects calcification by calcifiers inhabiting in macroalgal beds (Middelboe et al., 2007; Krause-Jensen et al., 2015, 2016; Duarte and Krause-Jensen, 2018; Wahl et al., 2018). However, there is limited field evidence for how the effects of macroalgal metabolism on the carbonate system extend to adjacent water bodies." We also added references about POC export in the Discussion section.

TECHNICAL CORRECTIONS Comment #28: l. 17-18: I suggest adding field measurements of carbon species to the set of applied methods.

Response: We agree with your comment.

Change: We have added "field measurements of carbon species" to the set of applied methods in Abstract as per your suggestion.

Comment #29: l. 22: Please change "offsite" to "offshore"

Response: We agree with your comment.

Change: We have changed "offsite" to "offshore" in Abstract.

Comment #30: l. 36: "is more labile" – I suggest rephrasing to underline the variable lability of macroalgal carbon.

Response: We agree with your comment.

Change: We have modified this sentence as follows: "Organic matter produced by macroalgae shows variable lability but are generally more labile than that produced by vascular plants"

Comment #31: l. 39: "is comparable to.." – or larger than?

Response: We agree with your comment.

Change: We have changed "is comparable to" to "larger than" according to the data shown in these references.

Comment #32: l. 40-41: "Macroalgal beds therefore have the potential to sequester substantial amounts of carbon in marine systems": This does not follow logically from the previous sentences – please rephrase.

Response: We agree with your comment.

Change: We have rephrased this sentence as follows: "Macroalgal beds therefore have the potential to regulate carbon dynamics in coastal ecosystems."

Comment #33: l. 49-50: please mention also the estimated contribution of DOC

to macroalgal C-sequestration in this previous study to highlight the hypothesis that macroalgal DOC is of major importance.

Response: We agree with your comment.

Change: We have added the estimated contribution of DOC to macroalgal C-sequestration (33%) in this sentence.

Comment #34: l. 67: "..this macroalgae is the dominant group in temperate regions": should it be e.g.". . .the Sargassaceae family of macroalgae is a dominant group in temperate regions"?

Response: We agree with your comment.

Change: We have rephrased this sentence as follows: "we focused on Sargassum beds because they are one of the dominant macroalgal habitats in temperate and tropical regions"

Comment #35: l. 74: "at a water depth.." –> "at water depths.."

Response: We agree with your comment.

Change: We have modified this wording as per your suggestion.

Comment #36: l. 188 and 193: Please add "(Table 1)" at the end of the sentence.

Response: We agree with your comment.

Change: We have added "(Table 1)" at the end of the sentences as per your suggestion.

Comment #37: l. 236: Please add after "(Table 2)" e.g.: ".., the rest being attributable to planktonic NCP."

Response: We agree with your comment.

Change: We have added modified this sentence as per your suggestion.

Comment #38: l. 241: please substitute "known" by "shown".

Response: We agree with your comment.

Change: We have changed "known" to "shown" in this sentence as per your suggestion.

Comment #39: l. 275/6: I suggest changing "The inhibition of macroalgal R by low water temperatures during the winter can explain the relatively high NCP values during the productive period at our study site (Table 1 and 2)." to "The inhibition of macroalgal R by low water temperatures during the productive winter can explain the relatively high NCP values observed at our study site (Table 1 and 2)."

Response: We agree with your comment.

Change: We have rephrased this sentence as per your suggestion.

Comment #40: - should "irradiation" be "irradiance"?

Response: We agree with your comment.

Change: We have changes "irradiation" to "irradiance" through the manuscript.
* * *

---

## Author Response (AR1)

**Author response to Associate editor's comments**

We thank you for your constructive comments. Below are the editor's comments and our responses to them. The original comments by the editor are in black font; they are followed by our responses in blue font. Line numbers in our responses are the line numbers of the revised manuscript. We have revised the manuscript title as reviewers' suggestion. We attached a marked-up manuscript version showing the changes made. Yellow highlighted points are the change points.

Thank you for posting replies to the referees comments. Please submit a revised version of your manuscript along the lines of your replies and also consider the following points:

- RMSE is the root mean square error (not "squared")

Response: We agree with your comment.

Change: We have modified the text per your suggestion. (L215)

- The same data should, in principle, not be presented twice (see Table 1 and Fig. 5).

Response: We agree with your comment.

Change: We have moved this table to the supplement (Table S2).

- In the revised manuscript, please clearly mention the limitations of the approach to measure NCC (uncertainty similar to the change) by providing some quantitative estimates of both.

Response: We agree with your comment.

Change: We have added an explanation of the limitations of the field bag approach to measure NCC. (L337)

Regards,

Jean-Pierre Gattuso

**Author response to RC1 by Albert Pessarrodona Silvestre**

We thank you for your constructive comments. Below are the reviewer's comments and our responses to them. The original comments by the editor are in black font, followed by our responses in blue. Line numbers in our responses are the numbers of the revised manuscript. We attached a marked-up manuscript version showing the changes made. Yellow highlighted points are the change points.

Comment #1: This is a great study that provides some highly valuable and relevant new insights about the potential transport of macroalgal carbon. Although the export of DOC below the mixed layer is believed to be the main pathway through which macroalgal carbon gets sequestered in the ocean, our understanding of the fate of macroalgal DOC after its release is very limited. This study presents tempting evidence of its potential export to offshore waters (but see some concerns below), which is an important step to verify the role of macroalgae in oceanic carbon sequestration. Overall, I found the study to be well conducted and well written. The authors provide a set of comprehensive measurements of different carbon compartments and forms, which I applaud. Although I am not familiar with some of the more technical protocols of the sample analysis, further reading and consulting suggest that they are standard.

Response: Thank you for your comments and suggestions. We have modified the manuscript after taking into consideration your suggestions. Please see details below.

Change: We have modified the manuscript after taking into consideration your suggestions. Please see details below.

Comment #2: One of my principal concerns is that the authors have not yet established a direct transport link between the water exported from the macroalgal bed and the waters at the offshore site. The authors found that (1) water near the macroalgal bed had different properties (namely: lower DIC, $fCO_2$ and higher DOC concentrations) than the water offshore except for February, when DOC concentrations were not significantly different. They then used mass balance models to simulate the diurnal changes in the carbonate and DOC system of the macroalgal bed (ln. 148); incorporating water exchange into their models helped better explain their readings (ln. 218, 245), which suggests that (2) there is water inflowing and outflowing at both the macroalgal bed and offshore site. There is however no direct demonstration that it is specifically the macroalgal bed water the one that reaches the offshore waters. This is a very important nuance, as the water that lowers the CO2 concentrations and enhances atmospheric CO2 uptake at the offshore site could come from other habitats that "produce" low DIC, high DOC water (e.g. seagrass meadow). Characterizing the DOC profile of both waters could help shed light on the provenance of that water.

Response: We agree with your comment. Our results show that low-CO₂ and high-DOC water is

exported from the macroalgal bed, but this finding does not directly demonstrate that water from the macroalgal bed reaches the offshore waters and affects carbon dynamics. We did not, however, find another nearby source of water with such a low $CO_2$ and high DOC content. We have thus rephrased the sentences about this claim throughout the manuscript. We have also added some explanations in the discussion about the need for a future study of the origin of bodies of water affected by coastal vegetation.

Change: We have rephrased the title and the sentences about this claim in the Abstract (L23), section 4.1 (L302), and section 5 (L422). We have said that macroalgal beds "can" create areas of adjacent water that serve as $CO_2$ sinks. We have also added some comments in the discussion about the need for a future study of the origin of bodies of water affected by coastal vegetation. (L412)

Comment #3: The mass balance models only consider changes due to processes related to macroalgal metabolism, but some could argue that they are missing some parameters. For example, volatile and semi-volatile compounds can be an important fraction of the DOC, and can be volatilize to the atmosphere (Ruiz-Halpern, Vaquer-Sunyer, & Duarte, 2014) instead of remaining in the water column as assumed here. Similarly, some of the other processes that can affect the DIC pools (e.g. dissolution, chemical addition; (Langdon et al., 2003)), are not considered. If the authors consider that those fluxes are negligible that is fine, but they should provide evidence to back their approach.

Response: In this study, the "DOC" did not contain a volatile fraction because we measured DOC as non-purgeable organic carbon according to a well-established method. Because the model calculated only non-purgeable DOC, the volatilization of DOC could be ignored. In the case of the DIC pools, carbonate dissolution and calcification were included in the mass balance model (Eq. 14).

Change: We have specified that our DOC was non-purgeable organic carbon. (L146)

Comment #4: It is very valuable that the study measurements were conducted at two separate time points albeit in the same season which gives an idea of the variability associated with the carbon flows estimated in the study. For instance, both the amount of DOCM and its constituents (as suggested by the different decomposition rates) were different across months (Wada et al., 2008). These points should be further elaborated to produce a rich and interesting discussion section. It would also be worth discussing how other species of macroalgae may differ in the production and characteristics of their DOC, as S. horneri was not the dominant species in the bed. Another limitation worth discussing is that DOC incubations for the degradation experiments were also maintained at a constant temperature (22), which may not necessarily reflect conditions in the field.

Response: The difference in the initial $DOC_M$ concentrations in the macroalgae bags between February and March experiments could have been caused by the differences in the biomasses of macroalgae and

volumes of water in the experimental bags. We have added a discussion about this point. We have added some discussion about seasonal and interspecific variations in the release rates of refractory DOC by referring to previous studies. In this study, we used room temperature (22°C), which is higher than the in situ temperature, for both study periods to compare the quality of organic matter. We have added a discussion about the effect of temperature on the microbial degradation of DOC in the discussion section.

Change: We have added some discussion about this point as follows: "The difference in the initial $DOC_M$ concentrations in the macroalgae bags between February and March may have been caused by the differences in the biomass of macroalgae and volume of water in the experimental bags (Fig. 6a, b). Variations of DOC concentrations may affect degradation rates via resource limitation of microbial activity (e.g., Arrieta et al., 2015). Understanding of the fate of macroalgal DOC would be enhanced by the assessment of the physical and biochemical factors that regulate microbial degradation of DOC." (L384) We have added some discussion about seasonal and interspecific variations in the release rates of refractory DOC as follows: "The phlorotannin contents of macroalgal thalli vary among seasons, growth phases, and species (Steinberg, 1989; Kamiya et al., 2010), and these variations may regulate seasonal and interspecific variations in the biological recalcitrance of macroalgal DOC." (L375) In this study, we used room temperature (22°C), which was higher than the in situ temperature for both study periods, to compare the quality of organic matter. We have added some discussion about the effect of temperature on the microbial degradation of DOC in the discussion section. (L381)

Comment #5: Finally, some of the sections of the manuscript also need to be further clarified, as it is difficult for the reader to grasp how some very key parameters where calculated. For example, it is unclear how the gross community production, respiration and calcification were calculated from the DOC bag experiments (ln. 160), all of which are key parameters in the model. It is also not very clear how the tidal water exchange (EXtide) rate was estimated from changes in depth (ln. 169)

Response: We agree with your comment.

Change: We have added relevant equations and explanations for the estimation of metabolic parameters in the Materials and methods section (section 2.4, L149). We have also added a schematic diagram of carbon pools and flows to clarify our mass balance model (Fig. 2 in the revised manuscript). We have added an equation to clarify the calculation of $EX_{tide}$ (L212) and a diagram to illustrate the temporal changes in $EX$ with water height in Fig. 3e, j.

Specific comments

Comment #6: Ln 33: Add "far" before "been"

Response: We agree with your comment.

Change: We have added "far" before "been" as per your suggestion. (L37)

Comment #7: Ln. 37 Add "more" before "efficiently"

Response: We agree with your comment.

Change: We have added "more" before "efficiently" as per your suggestion. (L41)

Comment #8: Ln 45: stored where? In the sediments, water column...? Also, consider citing here (Queirós et al., 2019), which provides an example of macroalgal-sediment connectivity.

Response: We agree with your comment.

Change: We have added "stored in sediments and the water column" and the citation "Queirós et al., 2019" as per your suggestion. (L49)

Comment #9: Ln. 52: I suggest making the topic sentence of the paragraph the fact that DOC is believed (at least according to (Krause-Jensen & Duarte, 2016)) to be the principal pathway of macroalgal carbon sequestration (although). This will highlight more the relevance of this study, as more empirical support is needed to demonstrate the assumptions of (Krause-Jensen & Duarte, 2016)

Response: We agree with your comment.

Change: We have made the topic sentence as follows: "The export and persistence of macroalgal dissolved organic carbon (DOC) has been proposed to be principal processes of macroalgal carbon sequestration, but more empirical support is needed to quantify this carbon flow." (L51)

Comment #10: Ln. 55: This paragraph feels a bit out of place here, you are talking about DOC and all of a sudden start talking about the carbonate system. Consider rearranging/rewriting.

Response: We agree with your comment.

Change: We have added the following sentence at the top of this paragraph: "Even though macroalgal beds perform a significant function by assimilating organic carbon, the chemical kinetics of the carbonate system in the water column could cause them to be net $CO_2$ emitters via air–water $CO_2$ exchange." (L61)

Comment #11: Ln. 61: The sentence gives the impression that the effects of macroalgal metabolism in their surrounding waters have not been studied, which is not the case (the authors provide plenty of examples). What is truly novel is examining its effects on other water bodies. I suggest deleting "both macroalgal beds and"

Response: We agree with your comment.

Change: We have modified this sentence as follows: "... there is limited field evidence for how the effects of macroalgal metabolism on the carbonate system extend to adjacent water bodies." (L71)

Comment #12: Ln 67: Sargassaceous algae sounds a bit strange to me, perhaps just use Sargassum beds? Sargassums are also commonly found in tropical regions (Fulton et al., 2019), so I would suggest changing for "we focused on Sargassum beds because they are one of the dominant macroalgal habitats in temperate and tropical regions).

Response: We agree with your comment.

Change: We have replaced "Sargassaceous" with "*Sargassum*" throughout the manuscript. Also, we have changed the sentence as per your suggestion. (L77)

Comment #13: Ln. 69 The issue of carbon sequestration was not directly addressed in this paper, as no evidence that the carbon measured is locked away from the atmosphere for very long periods of time (decades-centuries) is presented. Although some of the DOC did not decompose after 150 days under constant experimental conditions, it is not known how long it would remain in the field or whether it could reach the mixed layer. I suggest cutting similar claims made throughout the ms

Response: The elucidation of the mechanisms of long-term $CO_2$ sequestration by macroalgae is a final goal of our study, but the present study addressed the mechanism of $CO_2$ uptake by macroalgae. We have modified this point throughout the manuscript as per your suggestion.

Change: We have clarified that our research goal was to assess the key mechanisms of $CO_2$ uptake by the macroalgal bed in the Abstract and Introduction. (See also comment #2) (L24, L79)

Comment #14: Ln. 75. Given that the water inflowing and outflowing from the bed is so important for this study, the readers would appreciate more details about the water movements around the study area (e.g. tidal characteristics, exposure)

Response: We agree with your comment.

Change: We have added the following sentence as per your suggestion: "The study site is characterized by a relatively high tidal amplitude (<4 m), and it is adjacent to a deep strait (~60 m)." (L86)

Comment #15: Ln. 79. This sentence is a bit redundant from the one in Ln 76. Consider merging them.

Response: We agree with your comment.

Change: We have modified this sentence as follows: "Surface water samples for analyses of DIC, TAlk, and DOC were collected from a research vessel three times (10:00, 13:00, and 16:00) during the daytime (approximately from 7:00 to 17:00) during both surveys at five stations (H1–H5)." (L90)

Comment #16: Ln. 96. Is that the volume of seawater in the bag?

Response: Yes, this is the volume of seawater in each bag.

Change: We have added the words "in each bag" in line 110.

Comment #17: Ln. 109. Please indicate the pore size of the filter. Was the filtering pressurized?

Response: We agree with your comment.

Change: We have added the information about pore size (0.7 μm) and the filtering process ("under reduced pressure"). (L123)

Comment #18: Ln. 127. What concentration of KHPh?

Response: We have added the concentrations of potassium hydrogen phthalate (83, 166, and 332 μM).

Change: We have added the concentrations of potassium hydrogen phthalate. (L147)

Comment #19: Ln 140. At what height was the wind speed measured at Agenosho?

Response: The altitude was 6.5 m.

Change: We have added this information. (L182)

Comment #20: Ln 143. Delete "that"

Response: We agree with your comment.

Change: We have deleted "that" as per your suggestion (L185).

Comment #21: Ln. 148. Using the active voice is more readable in this instance. "We simulated the diurnal changes and budgets of the carbonate system and DOC in the macroalgal bed using mass balance models"

Response: We agree with your comment.

Change: We have modified the sentence as per your suggestion. (L190)

Comment #22: Ln. 151. This sentence seems to indicate that you changed the depth of the macroalgal bed. Please rewrite. Was the tide simulated by changing water height over the bed?

Response: We simulated the tide by changing water height over the bed.

Change: For clarification, we have modified the sentence as follows: "..., and the tide was simulated by changing the water height in synchrony with the observed tide." (L193)

Comment #23: Ln 152. The average Sargassum biomass used was derived from the field surveys, right? Please state so

Response: Yes, it is right.

Change: We have added the sentence "the average biomass of *Sargassum* algae obtained from the field survey". (L194)

Comment #24: Ln 157. The amount of formulas, acronyms and parameters used in the manuscript can

be a bit overwhelming. I encourage the authors to consider making a first figure with a schematic diagram of the different carbon pools and fluxes, as well as different carbon forms (e.g. POC, PIC, DOC, DIC...) and the processes that affect them (e.g. primary production, calcification...). That figure could include the formulas in lines 157-159 to show how they were calculated in the mass balance models. I think this could be very useful to the reader.

Response: We agree with your comment.

Change: We have added the new figure with a schematic diagram of the different carbon pools, fluxes, and processes (Fig. 2 in the revised manuscript).

Comment #25: Ln. 160. It is very unclear how all these parameters where calculated. Did you use some sort of relationship between DOC release and productivity? Please provide further details.

Response: We agree with your comment.

Change: We have added the equations and explanations for metabolic parameter estimation in the Materials and methods section. (section 2.4, L149)

Comment #26: Lns 165-166. They can be just one sentence

Response: We agree with your comment.

Change: We have modified the sentences as per your suggestion. (L205)

Comment #27: Lns 192-193. They can be just one sentence

Response: We agree with your comment.

Change: We have modified the sentences as per your suggestion. (L241)

Comment #28: Ln 205. The use of "g WW" is more standard. Also wet weight (WW) needs to be abbreviated somewhere in the paper.

Response: We agree with your comment.

Change: We have replaced "g-ww" with "g WW" throughout the manuscript and have noted the abbreviation in Materials and methods. (L118)

Comment #29: Ln. 208-209. Please provide statistical evidence that the decrease in time is statistically significant.

Response: We agree with your comment. We have conducted a Welch's two-sample $t$-test to detect the differences between the initial and final concentrations of DOC during degradation experiments. The conclusion has not been changed.

Change: We conducted a Welch's two-sample $t$-test to detect the differences between the initial and final concentrations of DOC during degradation experiments. We have described the analytical

process in the Materials and methods section (L229) and the results of this analysis in the Results section. (L257)

Comment #30: Ln 210. Perhaps it would be informative to include those final percentages in Fig. 4, as the decrease is a bit hard to observe in some panels (e.g. 4b)

Response: We agree with your comment.

Change: We have added these final percentages in Fig. 6 in the revised manuscript.

Comment #31: Ln. 218. Please provide an index of how well the model fits the data. This way you can say that a model improves or worsens by adding/removing water exchange.

Response: We have added the explanation for the model improvement (the change in the RMSEs of every parameter) by considering water exchange in this paragraph and the legend of Fig. 5. In the previous version of our manuscript, model fitting was performed by minimizing RMSEs solely for the DIC model, but this procedure ignores the errors in the other parameters (i.e., TAlk, DOC, and $fCO_2$). We have modified this model fitting method as follows: "The value of $EX_r$ was chosen so as to minimize the root mean square error (RMSE) of the modelled values versus the observed values. RMSEs were calculated for the $z$-scores of DIC, TAlk, DOC, and $fCO_2$, which were equated to the differences between the modelled values and the means of the observed values divided by the standard deviations of the observed values. The value of $EX_r$ that minimized the averaged RMSEs for these four parameters was determined for each survey." This modification has changed the results of the water exchange rate and carbon budgets, but the conclusion has not been changed.

Change: We have added the explanation for the model improvement (the change in the RMSEs of every parameter) by considering water exchange in this paragraph (L264) and the legend of Fig. 3 in the revised manuscript. We have modified the model fitting method (L215) and the related results (Fig. 3 and Table 2, 3) and discussion (L357).

Comment #32: Ln 238. Add "For example", before DIC uptake"

Response: We agree with your comment.

Change: We have added "For example," before DIC uptake" as per your suggestion. (L289)

Comment #33: Ln. 168: The estimation of water exchange is crucial for the aims of this paper. I am having a bit of trouble understanding how you EXtide was estimated from changes in depth. Is that referring to tidal height? It could be helpful if some example values are provided (e.g. is the number greater on spring tides, what is the maximum value it can attain? 1? What would that mean)

Response: We agree with your comment.

Change: We have added an equation for calculating $EX_{tide}$ for clarification (L212), and we have

included a figure that illustrates the temporal changes in *EX* with water height (Fig. 3e, j).

Comment #34: Ln. 256. I wonder how seasonality will affect the fate of the DOC released as well. How do oceanographic conditions vary in the study area?

Response: In this study, we did not collect seasonal data for macroalgal DOC and oceanographic conditions, but we have added some discussion about this point (temperature and oceanographic conditions).

Change: We have added some discussion about how the fate of macroalgal DOC depends on temperature and oceanographic conditions. (L381, L414)

Comment #35: Ln 274. You may also be interested in the extensive work of Sophie Martin in maerl beds e.g. (Martin, Clavier, Chauvaud, & Thouzeau, 2007)

Response: We appreciated this comment.

Change: We have added a citation to this reference and its NCP value in the revised manuscript. (L324)

Comment #36: Ln. 296-297. These two statements seem contradictory

Response: We agree with your comment.

Change: For clarification purposes, we have modified the sentence as follows: "Although the DOC release rates were similar between our two surveys, the percentages were very different between February (34 %) and March (6 %) (Fig. 7)." (L352)

Comment #37: Ln. 306. Very interesting find!

Response: Thank you!

Change: We have added some discussion about seasonal and interspecific variations in the release rates of refractory DOC by referring to previous work. (See also comment #4) (L375)

Comment #38: Ln. 320. Insert "considered as" before "are"

Response: We agree with your comment.

Change: We have modified this sentence as per your suggestion. (L394)

Comment #39: Ln. 321. Consider "[...] export of particulate macroalgal carbon (e.g. entire thalli and fragments) to the deep sea [...]"

Response: We agree with your comment.

Change: We have modified this sentence as per your suggestion. (L395)

Comment #40: Figure 4. Consider stating the percentage of DOC remaining in each of the treatments

of panels 4a and 4b as it is a big hard to tell how much remained sometimes. Also consider shading the area between the two treatments and indicating that it corresponds to the macroalgal DOC ($DOC_M$; ln. 121).

Response: We agree with your comment.

Change: We have added the final percentages of DOC in each treatment inside the panels in Fig. 6 in the revised manuscript. We have also shaded the area between the two treatments in Fig. 6 and indicated that it corresponds to the $DOC_M$.

Comment #41: Figure 5. I think that plotting the value of *EX* in this graphs would be very valuable, as it would help the reader understand what is the water doing (inflow or outflow), and how this affects the readings at the macroalgal bed and offshore sites. The mass balance model should also predict the observations at the offshore site; please plot those ones as well.

Response: We agree with your comment about *EX*. The mass balance model cannot predict the observations at the offshore site because the values of the offshore site were used as the boundary conditions for the inflowing water.

Change: We have added a plot of *EX* along with water height in Fig. 3e, j. We have added an explanation to clarify that the values at the offshore site were used as the boundary conditions for the inflowing water in the Materials and methods section. (L194)

Comment #42: Figure 6. I suggest putting a dashed line through the middle of the panes to clearly delineate the offshore waters from the macroalgal bed. Also, put the titles of "Offshore" and "Macroalgal bed" at the very top so it is easier to read. I think that using symbols instead of the photo of the macroalgal bed would declutter the figure and make it more understandable. For instance, the ones at https://ian.umces.edu/imagelibrary/displayimage-search-0-4529.html are freely available (with attribution) and make for very appealing figures.

Response: We agree with your comment.

Change: We have modified Fig. 7 as per your suggestion. We have included dashed lines to delineate the offshore waters from the macroalgal bed. We have also included the titles "Offshore" and "Macroalgal bed" at the top of Fig. 7. We have included sketches of macroalgae in Fig. 7.

**Author response to RC2 by Dorte Krause-Jensen**

We thank you for your constructive comments. Below are the reviewer's comments and our response to them. The original comments by the reviewer are in black font; these are followed by our responses in blue font. The line numbers in the our responses are the line numbers of the revised manuscript. We attached a marked-up manuscript version showing the changes made. Yellow highlighted points are the change points.

GENERAL COMMENTS

Comment #1: This study documents the exchange of dissolved carbon between a macroalgal habitat and adjacent waters. The study highlights that macroalgal metabolism and excretion of dissolved organic carbon (DOC) during the productive phase of the vegetation create low $CO_2$ concentration and high DOC concentration that, via water exchange, propagates from the macroalgal habitat to waters beyond the habitat. The low $CO_2$ concentrations created by macroalgae thereby contribute to increased air-sea $CO_2$ uptake both at the habitat and beyond, and export of DOC beyond the habitat suggests a potential of this carbon to reach oceanic sinks. These findings add significantly to the limited field evidence of the effect of macroalgal metabolism on dissolved carbon concentrations and the size of macroalgal-associated carbon fluxes and potentials for C-sequestration. Such evidence is important to underpin the recent notion that macroalgae contribute significantly to global C-sequestration, with the majority of the sequestration being supported by dissolved organic carbon reaching oceanic C-sink. The combination of in-situ measurements and flux studies, degradation experiments and modeling strengthen the findings. The study can be improved by adding detail in the method description, presentation and discussion of results and reference to earlier findings.

Response: Thank you for your comments and suggestions. We have modified the manuscript after taking into consideration your suggestions. Please see details below.

Change: We have modified the manuscript after taking into consideration your suggestions. Please see details below.

SPECIFIC COMMENTS

Methods

Comment #2: Field surveys (l. 76-80): - Please specify that the field studies were conducted during a diurnal cycle in February and March, respectively, and please underline the timing of sampling as well as the timing of sunrise/sunset so that the reader knows how the diurnal cycle was represented. Please also mention that February /March is the local winter time.

Response: Field surveys were conducted only during the daytime. To estimate diurnal cycles including nighttime, we used a field-bag method (especially respiration rate) and mass balance modelling (L208).

We collected water samples three times at 10:00, 13:00, and 16:00 during the daytime (approximately from 7:00 to 17:00).

Change: We have added this information in this paragraph. (L90) We have also noted that February and March are included in the local winter period in this paragraph. (L89)

Comment #3: Field bag experiments: - (l. 93) Was the ambient seawater for the macroalgal bags and control bags taken from the macroalgal site?

Response: Yes, the ambient seawater for every treatment was collected in the macroalgal bed.

Change: We have added this information in this paragraph. (L103, L106)

Comment #4: Biomass, cover and species composition (l. 100-106): - How long were the transect lines? While cover was assessed every 10 m (in 1 x 1 m quadrates) it is not clear how biomass assessments relate to cover assessments. Were the five 0.5 x 0.5 m biomass samples taken within quadrates assessed for cover and where cover estimates documented dominance by sargassaceous algae? Or were the biomass samples placed e.g. randomly within the belt dominated by sargassaceous algae? Please add detail.

Response: Both transect lines were 120 m. Five quadrats (0.5 m × 0.5 m) for quantifying biomass were randomly located in the area dominated by *Sargassum* algae along each transect.

Change: We have added this information in this paragraph. (L113, L117)

Comment #5: Degradation experiment - Why were the samples stored at room temperature (22°C)? Did this correspond to the in situ temperature? Were the samples aerated or did they turn anoxic during the incubation? How were degradation rates calculated?

Response: In this study, we used room temperature (22°C), which was higher than the in situ temperature, for both study periods to compare the quality of organic matter. The headspace in the bottles provided sufficient oxygen for aerobic degradation.

Change: We have added some discussion about the effect of temperature on the microbial degradation of DOC in the discussion section (L131, L381). We have added an explanation about the purpose of the headspace in this paragraph (L127). We have added an equation for calculating degradation rates (*k*) (L139).

Comment #6: Mass balance modeling - It is not entirely clear how this modeling was conducted - please expand the explanation. As far as I understand, the modeling was conducted solely for the macroalgal site (and not for the off shore site), please state this clearly. –

Response: As you say, this modelling was conducted solely for the macroalgal site, and the observed values of the offshore site were used as the boundary conditions for the carbon that inflowed to the

macroalgal bed.

Change: We added some sentences to clarify this point in this paragraph. (L194)

Comment #7: L. 154: Are the initial values for the macroalgal site estimated to be diurnal averages measured at the off shore site? (Please indicate in the text that the initial values are denoted "0" in the formula). (To clarify, I suggest moving the sentence (l. 163) "The values of DICO, TAlkO, and DOCO were the mean values at station H5." to follow l.154.)

Response: We agree with your suggestion.

Change: We have moved this sentence and rephrased it as follows: "The parameters $DIC_O$, $TAlk_O$, and $DOC_O$ in Eqs. (14–16) are the mean values of DIC, TAlk, and DOC, respectively, at the offshore station (H5), and the initial values in the simulation were equated to those values. Namely, $DIC_{(0)}$, $Talk_{(0)}$, and $DOC_{(0)}$ were equated to $DIC_O$, $TAlk_O$, and $DOC_O$, respectively." (L205)

Comment #8: L. 160-162: Please explain in more detail how the central parameters GCP, R and CC were determined (based on start/end and light/dark and macroalgal/control measurements) and which day length was applied. Regarding the calculation of calcification – please also see e.g. Wahl et al. 2018.

Response: We agree with your suggestion. We used the lengths of photoperiods explained in Line 167 and Table 2.

Change: We have added the equations and explanations for the estimation of metabolic parameters. (section 2.4, L149)

Comment #9: - Please make it clear in the methods section how the two different model scenarios (i.e. with and without considering water exchange, blue line and black line of Fig 5) were calculated.

Response: We agree with your suggestion.

Change: We have added an explanation of the two different model scenarios in this paragraph. (L220)

Results

Comment #10: (3.1) Carbonate system and DOC - Net community calcification: Did your study allow calculating potential differences in calcification between light and dark settings? - Fig. 2: Are the two lines significantly different?

Response: Because we used both transparent and dark bags for measuring net community calcification (NCC), we could calculate NCC rates in both light and dark settings (Table S3 in the Supplement). As we have pointed out in the previous manuscript, it is difficult to discuss the differences in NCC values between light and dark settings because the uncertainties of the NCC values derived from the measurement precision were comparable to the observed values. We did not conduct a statistical

analysis here because we did not intend to discuss the differences between March and February.

Change: For clarification, we have added an explanation about the limitation of quantifying NCC in this study. (L337)

Comment #11: (3.2) Biomass/cover - Fig. 3: Relationships between cover and biomass (relates to the question on how biomass samples were taken): How come that the highest biomass in panel a corresponds to the lowest coverage in panel b? And that the lowest biomass of sargassaceous algae (and highest relative biomass of "others" in panel c corresponds to a high (absolute and relative) cover of sargassaceous algae in panel d? Are there any significant biomass-cover relationships?

Response: As we noted in our reply to the question about how biomass samples were taken, quadrats for biomass and coverage were randomly located along each transect. We thus think that the heterogenous colonization of macroalgae (e.g., patches) caused the inconsistency. Because we used only the averaged biomass for the mass balance modelling, this inconsistency between biomass and coverage did not change the conclusion.

Change: We have added an explanation about how biomass samples were taken in the Materials and methods section. (L117)

Comment #12: (3.3) Degradation of DOC - l. 211-212 "Degradation rates (k) estimated by exponential fitting were 0.0044 d−1 and 0.0018 d−1 in February and March, respectively." Please clarify how degradation rates were calculated by e.g. chnagingt he sentence to "Degra-dation rates (k) estimated by exponential fitting of XXX vs XXX were 0.0044 d−1 and 0.0018 d−1 in February and March, respectively." and specify XXX.

Response: Thank you for your comment. We have added an explanation.

Change: We have added an explanation and equation for calculating degradation rates ($k$) in the Materials and methods section. (L137)

Comment #13: Fig. 4: - It is notable that DOC concentrations of the control bags were similar between Feb and Mar while DOC concentrations of macroalgal bags differed considerably, with final concentrations being about 140 μM in Feb and <100 μM in Mar. Please discuss this in the discussion section..

Response: The difference in the initial $DOC_M$ concentrations of the macroalgae bags between February and March could have been caused by the differences in the biomass of macroalgae and volume of water in the experimental bags. We added some discussion about this point.

Change: We have added discussion about this point as follows: "The difference in the initial $DOC_M$ concentrations in the macroalgae bags between February and March may have been caused by the differences in the biomass of macroalgae and volume of water in the experimental bags (Fig. 6a, b).

Variations of DOC concentrations may affect degradation rates via resource limitation of microbial activity (e.g., Arrieta et al., 2015). Understanding of the fate of macroalgal DOC would be enhanced by the assessment of the physical and biochemical factors that regulate microbial degradation of DOC." (L384)

Comment #14: - The 4th control sampling for March has high variability – might one sample be contaminated and should be omitted? - Panel c: Should the line-fit not be exponential?

Response: Thank you very much for your comment. We conducted triplicate analyses on each sample for DOC measurements using the TOC analyzer to reduce the analytical uncertainty, and we used the average of these triplicate measurements in our calculations. One of the samples for the 30-day control contained an outlier (i.e., erroneous value) among the triplicate values. This outlier caused anomalously high variability. In the revised manuscript, we have omitted this erroneous value and modified Fig. 4b, c and related sentences. We also rechecked every datum, and the other values were not erroneous. After this correction, plots became suitable for exponential fitting ($R^2$ was improved).

Change: We have omitted the erroneous value and modified Fig. 6b, c in the revised manuscript and related sentences. (L260, L378)

Comment #15: - How were degradation rates calculated. Based on fits of data in panel c? Why not based on fitting an exp decline curve to the macroalgal data of panel a and b?

Response: We have added an explanation and equation for calculating degradation rates ($k$) in the Materials and methods section. The focus of this degradation experiment was on the quantification of refractory DOC derived from macroalgae. We decided not to fit an exponential decay curve to the data from the macroalgal bags in Fig. 4a, b because the DOC in the macroalgal bags included ambient DOC.

Change: We have added an explanation and equation for calculating degradation rates ($k$) in the Materials and methods section. (L137)

Comment #16: (3.4) Carbon budgets - L. 214. "The mass balance models simulated the temporal changes in carbonate chemistry and DOC concentration (Fig. 5)." It is not clear how the mass balance models did this - please expand in the methods section and also elaborate a bit more here. Please explain that the model simulations represent both the situation when water exchange is taken into account (blue lines in Fig. 5) and the situation when it is not (black lines in Fig. 5). - I also suggest adding more detail to the legend of Fig. 5 to specify the significance of the blue line (in contrast to the black line), which is not mentioned in the current version of the legend.

Response: We have added an explanation of the two different model scenarios in the Materials and methods and result section. We have added a clarification of the model improvement (the change in

the RMSE of every parameter) due to consideration of water exchange in this paragraph and the legend of Fig. 5. In the previous version of our manuscript, model fitting was performed by minimizing RMSEs solely for the DIC model, but this procedure may lead to errors in other parameters (i.e., TAlk, DOC, and fCO$_2$). We have modified the description of the model fitting method as follows: "The value of $EX_r$ was chosen so as to minimize the root mean square error (RMSE) of the modelled values versus the observed values. RMSEs were calculated for the $z$-scores of DIC, TAlk, DOC, and fCO$_2$, which were equated to the differences between the modelled values and the means of the observed values divided by the standard deviations of the observed values. The value of $EX_r$ that minimized the averaged RMSEs for these four parameters was determined for each survey." This modification has changed the calculated water exchange rate and carbon budgets, but the conclusion has not been changed.

Change: We have added an explanation of the two different model scenarios in the Materials and methods (L220) and Results section (L263). We have added an explanation of the model improvement (the change in the RMSE of every parameter) achieved by considering water exchange in this paragraph (L264) and the legend of Fig. 3 in the revised manuscript. We have modified the model fitting method (L215) and the related results (Fig. 3 and Table 2, 3) and discussion (L357).

Comment #17: l. 216: How were the spans in hourly exchange rates calculated (35-48% and 50-76%) and why is the range not reported in Table 3 (35% and 50% is reported without any range).

Response: The $EX_r$ values were constants estimated for each survey, but $EX_{tide}$ values were variables that changed as a function of water height. For clarification, we have rephrased this sentence as follows: "Hourly water exchange rates (the sum of $EX_{tide}$ and $EX_r$) were . . .". We have shown the temporal change in water exchange rate (the sum of $EX_{tide}$ and $EX_r$) in Fig. 5.

Change: We have rephrased this sentence as follows: "Hourly water exchange rates (the sum of $EX_{tide}$ and $EX_r$) were . . ." (L268). We have shown the temporal change in water exchange rate (the sum of $EX_{tide}$ and $EX_r$) in Fig. 3e, j.

Comment #18: - It follows from the modeling approach (l. 223-224) that "DIC budgets driven by water exchange indicated a net input of DIC from offshore to the macroalgal bed (Fig. 6 and Table 3.)", - because otherwise the DIC levels at the macroalgal site would have been considerably lower that what was measured (as shown in Fig 5). However, the abstract says "The exported water lowered CO2 concentrations in the offsite surface water and enhanced atmospheric CO2 uptake.", and I think this statement needs be better underpinned by results and discussion.

Response: Thank you for your suggestion. We agree that our study did not directly demonstrate the enhancement of CO$_2$ uptake in the offshore site by the macroalgal bed.

Change: We rephrased the manuscript title and the sentence in the Abstract as follows: "These results

indicate that the exported water can potentially lower $CO_2$ concentrations in the offshore surface water and enhance atmospheric $CO_2$ uptake." (L23)

Discussion

Comment #19: - Net calcification (NCC): Please discuss /mention what may constitute the NCC: calcareous algae in the algal bed, mussels. . . ? Did you identify any variation in NCC between light and dark incubations? – please mention/discuss. For these discussions I suggest referring/comparing to e.g. Wahl et al 2018 & Duarte & Krause-Jensen 2018.

Response: As described above, we could calculate NCC rates in both light and dark settings (Table S3 in the Supplement). However, it is difficult to discuss quantitatively the differences in NCC values between light and dark settings because the uncertainties of the NCC rates derived from the measurement precision were comparable to the observed values. We also think that the quantitative comparison between our data and previous work is difficult. However, we have cited this previous work about the calcification in macroalgal beds.

Change: For clarification, we have added an explanation about the limitation of quantifying NCC in this study. (L337) We have cited this previous work about calcification in macroalgal beds in the Introduction. (L69)

Comment #20: - L. 276-8: Please elaborate a bit on this in relation to the differences observed between the Feb and Mar measurements. Regarding "growth phase", please mention that the study took place during the productive period.

Response: In the present study, both surveys were conducted during the productive period, but the averaged biomasses per individual *S. horneri* used for the field bag experiment differed (February, 353 g WW; March 260 g WW). This difference may indicate a difference in growth phase.

Change: We have added this explanation in this paragraph. (L329)

Comment #21: - L. 305-307: Regarding the comparation between DOC turnover times: Were the experimental conditions similar?

Response: Wada et al. (2008) calculated degradation rates for a 30-day incubation, which was shorter than the incubation in our study. We thus recalculated degradation rates for a 30-day incubation and compared the results with Wada et al. (2008). This recalculated result showed the same trend.

Change: We have modified this sentence based on this recalculation (L373). We have also added a sentence regarding the temporal change in the degradation rate during the DOC degradation by referring to a previous study. (L378)

Comment #22: - Please discuss reasons for the big difference in the residual concentrations of DOC

in the degradation of material from Feb and March.

Response: As described above, the difference between the residual $DOC_M$ concentrations in the macroalgae bags in February and March could have been caused by the differences in the initial $DOC_M$ concentrations.

Change: We have added some discussion about this point as follows: "The difference in the initial $DOC_M$ concentrations in the macroalgae bags between February and March may have been caused by the differences in the biomass of macroalgae and volume of water in the experimental bags (Fig. 6a, b). Variations of DOC concentrations may affect degradation rates via resource limitation of microbial activity (e.g., Arrieta et al., 2015). Understanding of the fate of macroalgal DOC would be enhanced by the assessment of the physical and biochemical factors that regulate microbial degradation of DOC." (L384)

Comment #23: - C-sequestration. L. 322-323: I suggest mentioning that a first-order assessment suggested that almost 70% of global macroalgal C-sequestration is attributable to DOC export to the deep sea (Krause-Jensen & Duarte 2016).

Response: We agree with your suggestion.

Change: We have mentioned this point in this paragraph as per your suggestion. (L399)

Comment #24: Figs/Tables (in addition to what is mentioned above)

Fig 5 legend: Please mention that details regarding rates are available in Table 1. I suggest moving Table 1 to supplementary material as the data are already presented in Fig 5.

Response: We agree with your comment.

Change: We have moved this table to the supplement (Table S2).

Comment #25: Fig 6 & Legend: Please change "Carbon flows.." to "Dissolved carbon flows.." Please add explanation of RDOCm. Unclear what is the difference between NDR and DOC. The legend says that (107) and (88) represents DOC flows; then why are the numbers at the DOC arrows different? Please mention how the data were generated (model, degradation exp, bag exp – maybe using different colors). Mention that the same calcification rates are reported both as calcification rates and as inorganic biomass growth. Mention if the NCP is the sum of macroalgal and planktonic NCP.. Mention that details regarding rates of C-metabolism are available in Table 2 and details on water exchange rates are available in Table 3.

Response: Because we also showed the air–water $CO_2$ gas exchange flux in this figure, we think that "Carbon flows..." is a better explanation. We agree with the other comments.

Change: We have modified the legend as follows: "Carbon flows and community metabolism (NCP, net community production; NCC, net community calcification; NDR, net DOC release) in the

macroalgal bed. NCP, NCC, and NDR were calculated using the results of field bag experiments (details are available in Table S1 in the Supplement). Biomass growth in terms of organic carbon (OC) was calculated by subtracting NDR from NCP. Biomass growth in terms of inorganic carbon (IC) was the same as NCC. DIC and DOC flows via water exchange were estimated by mass balance modelling (details are available in Table 3). Community metabolism, biomass growth, and DOC outflow indicate the sum of macroalgal and planktonic carbon flows. The carbon flows due solely to macroalgae are shown in parentheses. Carbon fluxes were calculated in units of millimoles per square metre of the surface area of the macroalgal bed per day. $RDOC_M$ indicates refractory DOC released by macroalgae."

References

Comment #26: - It would be appropriate to mention the pioneer study by Smith 1981 suggesting that lowering of CO2 concentrations by macroalgae leads to increased CO2 uptake and to highlight that the current study not only confirms that this is the case but also takes the finding further by documenting that the effect extends beyond the macroalgal habitat.

Response: We agree with your comment.

Change: We have added a citation and some discussion of the previous study by Smith (1981). (L68)

Comment #27: - Line 60-61: "However, the effects of macroalgal metabolism on the carbonate system in both macroalgal beds and adjacent water bodies have not been quantified": Please consider rephrasing to say e.g. that there is limited field evidence on this.. Earlier studies have documented effects of macroalgal metabolism on the carbonate system (e.g. Wahl et al. 2018, Middelboe et al. 2007, Krause-Jensen et al. 2015 & 2016, Duarte & Krause-Jensen 2018), and some of these provide evidence of how diurnal/temporal variations in macroalgal metabolism affect calcification. You may also want to mention that there are recent studies on particulate organic carbon (POC) fluxes from macroalgae (e.g. Filbee-Dexter et al. 2018, Pedersen et al. 2019, Queirós et al. 2019...) but less information on DOC fluxes despite the expected major importance of macroalgal DOC fluxes for carbon sequestration.

Response: We agree with your comment.

Change: We have rephrased this sentence as follows: "Indeed, some previous studies have shown that macroalgal beds act as sinks for atmospheric $CO_2$ (Delille et al., 2009; Ikawa and Oechel, 2015; Koweek et al., 2017) and contribute substantially to global carbon fluxes (Smith, 1981; Krause-Jensen and Duarte, 2016). Macroalgal metabolism regulates diurnal and temporal variations in carbonate chemistry and affects calcification by calcifiers in macroalgal beds (Middelboe and Hansen, 2007; Krause-Jensen et al., 2015, 2016; Duarte and Krause-Jensen, 2018; Wahl et al., 2018). However, there is limited field evidence for how the effects of macroalgal metabolism on the carbonate system extend to adjacent water bodies." (L67) We also added references about POC export in the Discussion section.

TECHNICAL CORRECTIONS

Comment #28: l. 17-18: I suggest adding field measurements of carbon species to the set of applied methods.

Response: We agree with your comment.

Change: We have added "field measurements of carbon species" to the set of applied methods in the Abstract as per your suggestion. (L18)

Comment #29: l. 22: Please change "offsite" to "offshore"

Response: We agree with your comment.

Change: We have changed "offsite" to "offshore" in the Abstract. (L22)

Comment #30: l. 36: "is more labile" – I suggest rephrasing to underline the variable lability of macroalgal carbon.

Response: We agree with your comment.

Change: We have modified this sentence as follows: "Organic matter produced by macroalgae shows variable lability but it is generally more labile than that produced by vascular plants" (L40)

Comment #31: l. 39: "is comparable to.." – or larger than?

Response: We agree with your comment.

Change: We have changed "is comparable to" to "larger than" according to the data shown in these references. (L44)

Comment #32: l. 40-41: "Macroalgal beds therefore have the potential to sequester substantial amounts of carbon in marine systems": This does not follow logically from the previous sentences – please rephrase.

Response: We agree with your comment.

Change: We have rephrased this sentence as follows: "Macroalgal beds therefore have the potential to regulate carbon dynamics in coastal ecosystems." (L45)

Comment #33: l. 49-50: please mention also the estimated contribution of DOC to macroalgal C-sequestration in this previous study to highlight the hypothesis that macroalgal DOC is of major importance.

Response: We agree with your comment.

Change: We have added the estimated contribution of DOC to macroalgal C-sequestration (33%) in

Comment #34: l. 67: "..this macroalgae is the dominant group in temperate regions": should it be e.g.". . .the Sargassaceae family of macroalgae is a dominant group in temperate regions"?
Response: We agree with your comment.
Change: We have rephrased this sentence as follows: "We focused on *Sargassum* beds because they are one of the dominant macroalgal habitats in both temperate and tropical regions" (L77)

Comment #35: l. 74: "at a water depth.." –> "at water depths.."
Response: We agree with your comment.
Change: We have modified this wording as per your suggestion. (L85)

Comment #36: l. 188 and 193: Please add "(Table 1)" at the end of the sentence.
Response: We agree with your comment.
Change: We have added "(Table S2)" at the end of the sentence as per your suggestion. We have moved this table to the supplement (Table S2) as per the suggestions of reviewer 1 and the editor.

Comment #37: l. 236: Please add after "(Table 2)" e.g.: ".., the rest being attributable to planktonic NCP."
Response: We agree with your comment.
Change: We have modified this sentence as per your suggestion. (L287)

Comment #38: l. 241: please substitute "known" by "shown".
Response: We agree with your comment.
Change: We have changed "known" to "shown" in this sentence as per your suggestion. (L292)

Comment #39: l. 275/6: I suggest changing "The inhibition of macroalgal R by low water temperatures during the winter can explain the relatively high NCP values during the productive period at our study site (Table 1 and 2)." to "The inhibition of macroalgal R by low water temperatures during the productive winter can explain the relatively high NCP values observed at our study site (Table 1 and 2)."
Response: We agree with your comment.
Change: We have rephrased this sentence as per your suggestion. (L325)

Comment #40: - should "irradiation" be "irradiance"?
Response: We agree with your comment.

Change: We have changes "irradiation" to "irradiance" throughout the manuscript.

---

## Referee Report (RR1)

**Comments to the authors**

The authors satisfactorily addressed most of my concerns. The new version of the manuscript features a different way of calculating EXr taking into account more parameters (i.e. the four modelled parameters rather than just DIC). The new figures are insightful, the manuscript has significantly improved in quality and once published will make a valuable contribution to the field.I only have a few minor suggestions/edits to improve the overall readability and clarity of the text.

One of the key and most novel results of the paper is that a relevant fraction of the NCP of macroalgae is exported to the offshore site (via DOC), and that a portion of that is refractory (ln 25, 27). This significant result is "buried" towards the end of the results (e.g. ln 277) and discussion (e.g. Ln 344) sections. Flipping the structure of the results and discussion (talking first about the most novel results and its implications for the CO2 sequestration of macroalgae) could give it more of a punch. This is just a suggestion up to the authors discretion.

Ln. 14. Evidence that macroalgae-derived carbon is locked away from the atmosphere for very long periods of time (decades-centuries) is still contentious, so I don't think the word "sequestration" is appropriate here. Perhaps carbon assimilation?

Ln. 17. Add "the" before "productive"

Ln. 18. Define here (a few words in a parenthesis would work) what a lateral carbon flow is, as a readers not familiar with that concept may not know what you are referring too.

Ln. 108. The field bag experiments were conducted during one day both in February and March right? This should be clear in the text.

Ln. 113. Should be [of the] "study macroalgal bed" instead of "macroalgae"

Ln. 153. I don't understand the first negative sign, shouldn't it just be ΔDIC – 0.5·ΔTalk? Also, for lns. 153-157 you have a variable expressed in $\mu$mol C $\cdot$ L$^{-1}$ $\cdot$ h$^{-1}$, so the equation should account for volume of the bag right? (i.e. divided by TB not just T). Finally, consider expressing your equations as $\frac{1}{2}$ΔTalk or ½ ΔTalk (same for other terms) as it is more clear what term of the equation is divided by what number

Ln. 121. Starting stating the aims of the paragraph will improve readability. Could try something like "To examine the degradation rates of macroalgal DOC, DOC samples were obtained …" . Also make it clear that you are interested in measuring microbial-driven DOC degradation.

Ln. 203. replace "from the results" for "from changes in DIC, TAlk, and DOC measured in the field bag experiments"

Ln. 214. Cite here some factors (examples) that may be driving EXr so the reader gets a better idea of what that term EXr represents. Could wind-driven water exchange be one of them?

Ln. 241. Consider changing the sentence to state that, on average, DOC was higher in the macroalgal bed during both sampling times, but differences between the bed and offshore site were only significant in March.

Ln 249. State that NCC was positive during both months (~11-21 mmol C m$^{-2}$ d$^{-1}$) and that "the average carbon fluxes due to NCC were one to two orders of magnitude lower than those derived from NCP"

Ln. 257. State somewhere in the paragraph that there was less decrease in March.

Ln. 258. Link this sentence with the first one in ln 259 by inserting "suggesting that" at the end of ln 258.

Ln. 265. Something along the lines of "the RMSEs for the best-fitting models considering water exchange (mean: February, 0.55; March, 0.86) were lower than those assuming water exchange was zero (mean: February, 3.85; March, 3.13; Table 2)" might make the sentence a bit more clear.

Ln 267. Replace "changed little" for "saw little to no improvement"

Ln 268. Are 39 and 43 the percentage relative to EX (i.e. what percentage of EX is due to EXr), or just its total value? As far as I gather from ln 268, it refers to its total value and that means that EXr was the main driver of EX right? If so, make it clear in the text.

Ln. 322. Perhaps it would be more interesting for comparison's sake to convert Randall et al. 2019 estimates to carbon values assuming a photosynthetic quotient of 1?

Ln. 332. I don't understand how the relative growth rates (% d$^{-1}$) values were obtained if the field bag experiments were conducted only during one day. Please clarify.

Ln. 351-353. Collate in just one sentence.

Ln. 372. Should be "30-day"

Ln. 375. Start sentence with "Recalcitrant macroalgae compounds such as phlorotannins vary …"

Ln. 382. Add "microbial" before "degradation" to make clear that you only measured microbial-driven degradation of DOC under controlled conditions. Add a sentence somewhere in the paragraph stating that other factors (e.g. photochemical degradation) not measured in the study could also be important in driving DOC degradation. Also, change "should be overestimated" for "are potentially overestimates".

Ln. 396 Pessarrodona et al. 2018 and Pedersen et al. 2019 did not explicitly consider/measure POC export from macroalgal beds (rather POC production or release), so perhaps consider removing them.

Ln 400. Main pathway of macroalgal DOC sequestration is thought to be export below the mixed layer (rather than "deep sea")

Ln. 402. Perhaps saying that "the maximum residence time of dissolved mater in the study's oceanographic basin is approximately between 100-200 (depending on the season)" would make a stronger case for the potential export of your macroalgal RDOC outside the Seto Inland Sea.

Ln. 421. Delete "to the surrounding water"

Figure 2. This is a great Figure!

Figure 3. Perhaps labelling February and March on top of the left and right columns respectively could help with readability.

Figure 4. The relationships have an $R^2$ of 1, but the plots show some deviance of the predicted relationship (i.e. data points away from the regression line) in March for instance. The $R^2$ seems a bit high for empirical data?

Figure 7. The sentence "The carbon flows due solely […]" should go after the first sentence in the caption to inform the reader what the parentheses mean.

Table S2. Add "measured" in front of "in the surface"

---

## Author Response (AR2)

**Author response to Associate editor's comments**

We thank you for your constructive comments. Below are the editor's comments and our responses to them. The original comments by the editor are in black font; they are followed by our responses in blue font. Line numbers in our responses are the line numbers of the revised manuscript. We have attached a marked-up manuscript that shows the changes we have made highlighted in yellow.

We have implemented almost all the suggestions made by the Editor and reviewer. The following suggestions were not implemented:

1) The reviewer was confused by the minus sign added at the beginning of the NCP equations (Comment #8 by the reviewer), and you also suggested that we use the absolute value at that point Comment #5 by the editor). However, an absolute value would be inappropriate, because NCP can be negative when respiration exceeds GCP. We therefore think that it is reasonable to put a minus sign at the beginning of the equation, but for clarification we have rearranged the equation. We apologize that there were mistakes in other equations and have modified those equations. Because of those modifications, NCP rates and model output values changed by small amounts, but the discussion and conclusions were not changed.

2) The editor commented (Comment #7) that using different equations for the control and macroalgal metabolic parameters was a waste of space. However, we think that we should show both equations because macroalgal metabolic parameters have to be calculated by using control metabolic parameters to remove the effects of planktonic metabolism in the bags with macroalgae. To clarify this point, we have rearranged the equations.

3) The reviewer commented (Comment #25) that two of the references (Pessarrodona et al. 2018 and Pedersen et al. 2019) did not explicitly consider/measure POC export from macroalgal beds (rather POC production or release) and that we should perhaps consider removing them. However, we think that these referenced works are important because they show that macroalgal beds release large amounts of POC outside of the beds. We have removed the words "to depths below the mixed layer" and modified the sentence as follows: "The release and subsequent export of particulate macroalgal carbon (e.g., entire thalli and fragments) via physical processes would contribute to $CO_2$ sequestration" (L406)

[Comments before 2nd round review]

Dear Author,

Thank you for submitting a revised version of your manuscript submitted to Biogeosciences. I decided

to send it to referee #1 for another round of review. I will let you know the outcome as soon as possible.

Comment #1: I am not certain that the title adequately reflects the content of the manuscript. It is good that you added "can" to tone done the statement as only one bed was investigated and for only 2 days in the same season. Is "extended" justified? It is pretty strong especially considering the fact that no quantitative estimate of the particulate plus dissolved export is provided. Also, no estimate of the sink is provided. Perhaps "Macroalgal metabolism and lateral carbon flow can create significant carbon sinks". Here the emphasis would be on carbon (particulate and dissolved) rather than on CO2 which is only part of your story. Also "extended" would be toned down by using "significant". Just a suggestion to stimulate a revision.

Response: We agree with your comment. We think that the title "Macroalgal metabolism and lateral carbon flows can create significant carbon sinks" is better because this title tones down the statement and includes DOC dynamics per your suggestion.

Change: We have changed the title to "Macroalgal metabolism and lateral carbon flows can create significant carbon sinks".

Comment #2: I forgot to mention that Biogeosciences strongly promotes the full availability of the data sets reported in the papers that it publishes in order to facilitate future data comparison and compilation as well as meta-analysis. The availability of data by request to the authors is not satisfactory. You could upload the data sets in an existing database and providing the link(s) in the paper. Alternatively, the data sets can be published, for free, alongside the paper as supplementary information. The ascii (or text) format is preferred for data and any format can be handled for movies, animations etc…

Response: We agree with your comment.

Change: We have uploaded the dataset to the online database Zenodo. We have cited this database in the Data Availability section (L437).

[Comments after 2nd round review]

Comments to the Author:

Dear Author,

Thank you for submitting a revised version of your manuscript under consideration in Biogeosciences. It could be accepted for publication after minor revision (see below).

Comment #3: When submitting the revised version, please let me know which of the changes were not implemented, if any, and why. This would speed up final acceptance.

I look forward to hearing from you.

Best regards,

Jean-Pierre Gattuso
* * *
Response: We have implemented almost all the suggestions made by the Editor and reviewer. The suggestions that were not implemented were showed at the beginning of this letter.

Comment #4: - see comments provided in a pdf file by referee #1. Let me know if this file is not available to you.

- Section 2.4: I agree with referee #1. There are problems with the equations.

Response: Thank you for your suggestions.

Change: We have modified some (but not all) of the equations. Please see details below.

Comment #5: - Referee #1 is confused with the minus sign added at the beginning of the NCP equations. I assume this sign is there to make NCP positive. This is not the way to do it and I recommend adding vertical bars to indicate that the absolute value is taken. Like: |xxxxx|

Response: Because $\Delta$DIC was calculated as the final minus initial concentrations, $\Delta$DIC became negative under autotrophic conditions. NCP under autotrophic conditions has generally been represented as a positive value in previous publications. However, NCP should not be considered to be an absolute value, because it can be negative when respiration exceeds GCP. We therefore think that it is reasonable to place a minus sign at the beginning of the equation. We apologize that there were mistakes in other equations. We have modified these equations. Because of those modifications, NCP rates and model output values changed by small amounts, but the discussion and conclusion were not changed.

Change: We have continued to place a minus sign at the beginning of the NCP equations, but for clarification, we have rearranged the equations. We have corrected the mistakes in the equations. In Eqs 5 and 10, we have added a minus sign at the beginning because CC is generally represented as a positive value in previous studies. In Eqs 5, 6, 10, and 11, we have modified the equations and added some explanation (L170). Based on these modifications, values of metabolic parameters and model output values were changed by small amounts (L25, 27, 252, 253, 257, 270, 275, 285, 300, 311, 380, 392, 399; Fig. 3, 7; Table 1, 2, 3).

Comment #6: - I suggest to use "l" and "d" for light and dark.

Response: We agree with your comment.

Change: We have modified the explanations and equations in accord with your suggestion. (L168)

Comment #7: - Using different equations for the control and macroalgal metabolic parameters is a waste of space. Why not use the same equation like NCP = |(ΔDIC_l - 1/2 x ΔTALK_l)| / T. And add below the set of equations "ΔDIC, ΔTAlk, and ΔDOC were calculated as the final minus initial concentrations in the control and macroalgal experiments."

Response: Because macroalgal metabolic parameters must be calculated by using control metabolic parameters, we think that we should show both equations for clarification.

Change: We have not made the change you suggested, but we have rearranged the equations to clarify them. We have modified the sentence in accord with your suggestion (L167).

Comment #8: - It is not a good idea to use multiple "/" signs such as "x V / B / T". It should rather be " x V / (B x T)". Is that right?

Response: We agree with your comment.

Change: We have modified the equations in accord with your suggestion. (Eq. 7–11)

Comment #9: - 168-169 and 180-181: The source of the photoperiod and wind data should be provided: either a reference or the url of a web site.

Response: We agree with your comment.

Change: We have added the url. (L175, L188)

Comment #10: - 173: K x S x (...)

Response: We agree with your comment.

Change: We have modified the equations in accord with your suggestion.

Comment #11: - 178: 0.39 x U

Response: We agree with your comment.

Change: We have modified the equations in accord with your suggestion.

Comment #12: - 185: I assume you did not do the calculation of the carbonate chemistry with a calculator but with a software. Which one?

Response: We agree with your comment. We used the CO2SYS program.

Change: We have added an explanation as follows: "The values of $fCO_{2water}$ were estimated with the CO2SYS program (Lewis and Wallace, 1998) and ..." (L191).

**Author response to RC1 by Albert Pessarrodona Silvestre**

We thank you for your constructive comments. Below are the reviewer's comments and our responses to them. The original comments by the reviewer are in black font, followed by our responses in blue. Line numbers in our responses are the line numbers of the revised manuscript. We have attached a marked-up manuscript that shows the changes that we have made highlighted in yellow. We have changed the title in accord with the editor's suggestion.

Comment #1: The authors satisfactorily addressed most of my concerns. The new version of the manuscript features a different way of calculating $EX_r$ taking into account more parameters (i.e. the four modelled parameters rather than just DIC). The new figures are insightful, the manuscript has significantly improved in quality and once published will make a valuable contribution to the field. I only have a few minor suggestions/edits to improve the overall readability and clarity of the text.

Response: Thank you for your comments and suggestions.

Change: We have modified the manuscript after taking into consideration your suggestions. Please see details below.

Comment #2: One of the key and most novel results of the paper is that a relevant fraction of the NCP of macroalgae is exported to the offshore site (via DOC), and that a portion of that is refractory (ln 25, 27). This significant result is "buried" towards the end of the results (e.g. ln 277) and discussion (e.g. Ln 344) sections. Flipping the structure of the results and discussion (talking first about the most novel results and its implications for the CO2 sequestration of macroalgae) could give it more of a punch. This is just a suggestion up to the authors discretion.

Response: Thank you for your suggestion. We agree with your comment.

Change: We have moved the sentences about the DOC budget to the beginning of the paragraph in the Results (L282). In addition, we have moved the section "Refractory DOC release by macroalgae" to the beginning of the Discussion (i.e., section 4.1) (L291).

Comment #3: Ln. 14. Evidence that macroalgae-derived carbon is locked away from the atmosphere for very long periods of time (decades-centuries) is still contentious, so I don't think the word "sequestration" is appropriate here. Perhaps carbon assimilation?

Response: We agree with your comment.

Change: We have modified this sentence as follows: "important pathways for $CO_2$ uptake by macroalgal beds" (L14).

Comment #4: Ln. 17. Add "the" before "productive"

Response: We agree with your comment.

Change: We have added "the" before "productive" (L16).

Comment #5: Ln. 18. Define here (a few words in a parenthesis would work) what a lateral carbon flow is, as a readers not familiar with that concept may not know what you are referring too.

Response: We agree with your comment.

Change: We have added the explanation as follows: "lateral carbon flows (i.e., carbon exchanges between the macroalgal bed and the offshore)" (L18).

Comment #6: Ln. 108. The field bag experiments were conducted during one day both in February and March right? This should be clear in the text.

Response: Yes, your are right. We agree with your comment.

Change: We have added the words "during one day in both February and March of 2019" in L102:

Comment #7: Ln. 113. Should be [of the] "study macroalgal bed" instead of "macroalgae"

Response: We agree with your comment.

Change: We have modified the sentence in accord with your suggestion. (L115)

Comment #8: Ln. 153. (1) I don't understand the first negative sign, shouldn't it just be ΔDIC – 0.5·ΔTalk? (2) Also, for lns. 153-157 you have a variable expressed in μmol C · L-1 · h-1, so the equation should account for volume of the bag right? (i.e. divided by TB not just T). (3) Finally, consider expressing your equations as 12ΔTalk or ½ ΔTalk (same for other terms) as it is more clear what term of the equation is divided by what number

Response: (1) Because ΔDIC was calculated as the final minus initial concentrations, ΔDIC became negative under autotrophic conditions. NCP under autotrophic condition is generally represented as a positive value in previous studies. NCP is not an absolute value, because it can be negative when respiration exceeds GCP. We therefore think that it is reasonable to place a minus sign at the beginning of the equation. We apologize that there were mistakes in other equations. We have modified those equations. Because of these modifications, NCP rates and model output values changed by small amounts, but the discussion and conclusions were not changed. (2) In Eqs 2–6, we calculated the change in the concentrations due to phytoplankton in bags. The volume of the bag was not used in those equations. (3) Because we have changed the equations to fractional expressions for clarification purposes, we have expressed 1/2 as 0.5 for clarification.

Change: (1) We did not change the minus sign at the beginning of the NCP equations, but for clarification we have rearranged the equations. We have corrected the mistakes in the equations. In Eq 5 and 10, we have added a minus sign at the beginning because CC is generally represented as a

positive value in previous studies. In Eqs 5, 6, 10, and 11, we have modified the equations and added an explanation (L170). Because of this modification, metabolic parameters and model output values changed by small amounts (L25, 27, 252, 253, 257, 270, 275, 285, 300, 311, 380, 392, 399; Fig. 3, 7; Table 1, 2, 3). (2) We have made no change with respect to the volume. (3) We have changed the equations to fractional form to clarify them and expressed 1/2 as 0.5.

Comment #9: Ln. 121. Starting stating the aims of the paragraph will improve readability. Could try something like "To examine the degradation rates of macroalgal DOC, DOC samples were obtained …" . Also make it clear that you are interested in measuring microbial-driven DOC degradation.
Response: We agree with your comment.
Change: We have added the sentence "To quantify the degradation rates of macroalgal DOC due to microbial activity and to estimate the refractory fraction of that DOC, ..." in L124.

Comment #10: Ln. 203. replace "from the results" for "from changes in DIC, TAlk, and DOC measured in the field bag experiments"
Response: We agree with your comment.
Change: We have modified the sentence in accord with your suggestion. (L209)

Comment #11: Ln. 214. Cite here some factors (examples) that may be driving $EX_r$ so the reader gets a better idea of what that term $EX_r$ represents. Could wind-driven water exchange be one of them?
Response: We agree with your comment.
Change: We have added examples as follows "e.g., wind-driven water exchange and coastal currents". (L221)

Comment #12: Ln. 241. Consider changing the sentence to state that, on average, DOC was higher in the macroalgal bed during both sampling times, but differences between the bed and offshore site were only significant in March.
Response: We agree with your comment.
Change: We have added the sentence "On average, the DOC concentrations were higher in the macroalgal bed than at the offshore site, but the difference between them was significant only in March" in L247.

Comment #13: Ln 249. State that NCC was positive during both months (~11-21 mmol C m-2 d-1) and that "the average carbon fluxes due to NCC were one to two orders of magnitude lower than those derived from NCP"
Response: We agree with your comment.

Change: We have modified the sentence as follows: "The net community calcification (NCC) of macroalgae was positive during both months (11–21 mmol-C m$^{-2}$ d$^{-1}$), but the average carbon fluxes due to NCC were one to two orders of magnitude lower than those associated with NCP." (L253)

Comment #14: Ln. 257. State somewhere in the paragraph that there was less decrease in March.
Response: We agree with your comment.
Change: We have modified the sentence as follows: "The degradation rate ($k$) for 150-day incubations was higher in February (0.0044 d$^{-1}$) than in March (0.0021 d$^{-1}$)." (L266)

Comment #15: Ln. 258. Link this sentence with the first one in ln 259 by inserting "suggesting that" at the end of ln 258.
Response: We agree with your comment.
Change: We have modified the sentence as follows: "In contrast, the stability of DOC concentrations collected from control bags during the experiments ($p > 0.05$) suggested that ...". (L264)

Comment #16: Ln. 265. Something along the lines of "the RMSEs for the best-fitting models considering water exchange (mean: February, 0.55; March, 0.86) were lower than those assuming water exchange was zero (mean: February, 3.85; March, 3.13; Table 2)" might make the sentence a bit more clear.
Response: We agree with your comment.
Change: We have modified the sentence in accord with your suggestion. (L270)

Comment #17: Ln 267. Replace "changed little" for "saw little to no improvement"
Response: We agree with your comment.
Change: We have modified the sentence as follows: "those of DOC and TAlk showed little or no improvement". (L274)

Comment #18: Ln 268. Are 39 and 43 the percentage relative to EX (i.e. what percentage of EX is due to EXr), or just its total value? As far as I gather from ln 268, it refers to its total value and that means that EXr was the main driver of EX right? If so, make it clear in the text.
Response: Yes, they are relative to its total value.
Change: We have modified the sentences as follows: "The $EX_r$ rates were the main components of the hourly water exchange rates (the sums of $EX_{tide}$ and $EX_r$), which were estimated to be 39–52 % and 42–68 % in February and March, respectively (Fig. 3 and Table 3)." (L275)

Comment #19: Ln. 322. Perhaps it would be more interesting for comparison's sake to convert Randall

et al. 2019 estimates to carbon values assuming a photosynthetic quotient of 1?

Response: We agree with your comment.

Change: We have modified the sentence in accord with your suggestion. (L382)

Comment #20: Ln. 332. I don't understand how the relative growth rates (% d-1) values were obtained if the field bag experiments were conducted only during one day. Please clarify.

Response: Thank you for your comment. As we mentioned in the Materials and methods section, we estimated the daily metabolic parameters such as NCP by using the lengths of the photoperiods and the results of both transparent and dark bags. To clarify this point, we have modified the sentence.

Change: We have modified the sentence as follows: "The metabolic parameters were converted to daily areal rates (mmol-C $m^{-2}$ $d^{-1}$) by using the mean macroalgal biomass, the mean water depth, the lengths of the photoperiods, and the results of both daytime and night-time experiments." (L171)

Comment #21: Ln. 351-353. Collate in just one sentence.

Response: We agree with your comment.

Change: We have modified the sentence as follows: "Our results showed that Sargassum algae sometimes release a similar percentage of production as DOC (February, 35 %; March; 6 %), and the percentages were very different between the two months, despite the similarity of the DOC release rates (Fig. 7)." (L299)

Comment #22: Ln. 372. Should be "30-day"

Response: We agree with your comment.

Change: We have modified this text in accord with your suggestion. (L320)

Comment #23: Ln. 375. Start sentence with "Recalcitrant macroalgae compounds such as phlorotannins vary …"

Response: We agree with your comment.

Change: We have modified the text in accord with your suggestion. (L323)

Comment #24: Ln. 382. Add "microbial" before "degradation" to make clear that you only measured microbial-driven degradation of DOC under controlled conditions. Add a sentence somewhere in the paragraph stating that other factors (e.g. photochemical degradation) not measured in the study could also be important in driving DOC degradation. Also, change "should be overestimated" for "are potentially overestimates".

Response: We agree with your comment.

Change: We have modified the sentences in accord with your suggestion (L330, L331). We have added

the new reference and the sentence "The rates of DOC degradation processes, which were not measured in this study (e.g., photochemical degradation), might also be important in driving macroalgal DOC degradation (Wada et al., 2015)." (L337)

Comment #25: Ln. 396 Pessarrodona et al. 2018 and Pedersen et al. 2019 did not explicitly consider/measure POC export from macroalgal beds (rather POC production or release), so perhaps consider removing them.

Response: We think that these referenced works are important to show that macroalgal beds release large amount of POC to the outside of the beds. We have modified the sentence to clarify it.

Change: We have removed the words "to depths below the mixed layer" and modified the sentence as follows: "The release and subsequent export of particulate macroalgal carbon (e.g., entire thalli and fragments) via physical processes would contribute to $CO_2$ sequestration" (L406)

Comment #26: Ln 400. Main pathway of macroalgal DOC sequestration is thought to be export below the mixed layer (rather than "deep sea")

Response: We agree with your comment.

Change: We have modified the sentences in accord with your suggestion. (L411, L415)

Comment #27: Ln. 402. Perhaps saying that "the maximum residence time of dissolved mater in the study's oceanographic basin is approximately between 100-200 (depending on the season)" would make a stronger case for the potential export of your macroalgal RDOC outside the Seto Inland Sea.

Response: We agree with your comment.

Change: We have modified the sentence as "The maximum residence time of dissolved matter in the study's oceanographic basin is between 95–218 days depending on the season (Balotro et al., 2002), indicating that macroalgal RDOC can be exported to the outside of the Seto Inland Sea and to depths below the mixed layer via vertical mixing." (L413)

Comment #28: Ln. 421. Delete "to the surrounding water"

Response: We agree with your comment.

Change: We have modified the sentences in accord with your suggestion. (L433)

Comment #29: Figure 2. This is a great Figure!

Response: Thank you for your first-round review comment.

Change: No change.

Comment #30: Figure 3. Perhaps labelling February and March on top of the left and right columns

respectively could help with readability.

Response: We agree with your comment.

Change: We have modified the figure in accord with your suggestion. (Fig. 3)

Comment #31: Figure 4. The relationships have an R2 of 1, but the plots show some deviance of the predicted relationship (i.e. data points away from the regression line) in March for instance. The R2 seems a bit high for empirical data?

Response: $R^2 = 0.996$ in February and 0.995 in March.

Change: We have increased the number of decimal places. (Fig. 4)

Comment #32: Figure 7. The sentence "The carbon flows due solely […]" should go after the first sentence in the caption to inform the reader what the parentheses mean.

Response: We agree with your comment.

Change: We have modified the figure legend in accord with your suggestion. (Fig. 7)

Comment #33: Table S2. Add "measured" in front of "in the surface"

Response: We agree with your comment.

Change: We have modified the table caption in accord with your suggestion. (Table S2 in the Supplement)